# Pneumococcal and Influenza Vaccination Rates and Pneumococcal Invasive Disease Rates Set Geographical and Ethnic Population Susceptibility to Serious COVID-19 Cases and Deaths

**DOI:** 10.3390/vaccines9050474

**Published:** 2021-05-08

**Authors:** Robert Root-Bernstein

**Affiliations:** Department of Physiology, 567 Wilson Road, Room 1104 Biomedical and Physical Sciences Building, Michigan State University, East Lansing, MI 48824, USA; rootbern@msu.edu

**Keywords:** COVID-19, SARS-CoV-2, invasive pneumococcal disease, lower respiratory infections, pneumococcal vaccines, PCV13, PPV23, influenza vaccine, measles-mumps-rubella vaccine, BCG

## Abstract

This study examines the relationship of pneumococcal vaccination rates, influenza, measles-mumps-rubella (MMR) diphtheria-tetanus-pertussis vaccinations (DTP), polio, Haemophilus influenzae type B (Hib), and Bacillus Calmette–Guerin (tuberculosis) vaccination rates to COVID-19 case and death rates for 51 nations that have high rates of COVID-19 testing and for which nearly complete childhood, at-risk adult and elderly pneumococcal vaccination data were available. The study is unique in a large number of nations examined, the range of vaccine controls, in testing effects of combinations of vaccinations, and in examining the relationship of COVID-19 and vaccination rates to invasive pneumococcal disease (IPD). Analysis of Italian regions and the states of the United States were also performed. Significant positive correlations were found between IPD (but not lower respiratory infections) and COVID-19 rates, while significant negative correlations were found between pneumococcal vaccination and COVID-19 rates. Influenza and MMR vaccination rates were negatively correlated with lower respiratory infection (LRI) rates and may synergize with pneumococcal vaccination rates to protect against COVID-19. Pneumococcal and influenza vaccination rates were independent of other vaccination rates. These results suggest that endemic rates of bacterial pneumonias, for which pneumococci are a sentinel, may set regional and national susceptibility to severe COVID-19 disease and death.

## 1. Introduction

The ongoing COVID-19 pandemic poses two outstanding problems. One is why the symptoms associated with SARS-CoV-2 infection vary in individuals from unnoticeable to requiring intensive care. More specifically, a third of polymerase chain reaction-positive SARS-CoV-2-infected people are symptom-free; most experience mild-to-moderate symptoms that are treatable on an out-patient basis, while only 2 to 9 percent develop acute respiratory distress syndrome concomitant with clearly elevated cytokine levels that require hospitalization and intensive care unit treatment [1,2,3]. Secondly, why do national death rates from COVID-19 vary from less than 1 per million to thousands per million? [4] Logically, these two questions would seem to be related since the population risk of death is a function of the number of people who become seriously ill with COVID-19.

While age, population density, access to high-quality healthcare, and national and local public health mandates regarding mask use, social distancing, and COVID-19 testing rates undoubtedly account for some of the variances in both morbidity and mortality, a principle expounded by Louis Paster when he first stated the germ theory of disease is also relevant: the terrain is as important as the germ. The effects of a new infection vary according to the health of the individual, and the susceptibility of a population to a new infection is dependent on its overall, preexisting disease burden (including health factors such as nutrition). In this context, significant research has been devoted to the possibility that various non-SARS-CoV-2 vaccines may affect the propensity of individuals and of populations to contract COVID-19 and to die from its effects. This paper tests the particular possibility that vaccination rates and endemic bacterial disease burden significantly influence individual risks of contracting severe COVID-19 and therefore affect COVID-19 case and death rates.

The specific analyses that are undertaken here are as follows. The Introduction begins with a brief review of the incidence of various bacterial, fungal and viral infections known to co-exist with SARS-CoV-2 in severely affected COVID-19 patients and distinguishing these patients from those with asymptomatic and mild-to-moderate COVID-19. This section sets up the basic argument to be explored here that COVID-19 risk is associated with the broader disease burden experienced by individuals and populations. The next section then develops an important implication of the disease burden hypothesis, which is that some vaccinations, by lowering COVID-19-predisposing disease burden, may also decrease the risk of contracting and dying from COVID-19 itself. The current literature regarding COVID-19 risks in relation to vaccination rates with influenza, measles-mumps-rubella (MMR) diphtheria-tetanus-pertussis (DTP), polio, *Haemophilus influenzae* type B (Hib), and Bacillus Calmette–Guerin (tuberculosis) are therefore briefly reviewed. These introductory review sections establish that the severity of COVID-19 disease is related to the incidence of bacterial, fungal, and viral coinfections and that some vaccinations, particularly with pneumococcal and influenza vaccines, appear to provide protection against either acquisition of SARS-CoV-2 and/or severity of resulting COVID-19 disease.

One problem with many of the existing studies of possible COVID-19 protection provided by vaccinations is the lack of appropriate controls and/or use of inappropriate population data (such as that from nations that have extremely low rates of SARS-CoV-2 testing, do not report some vaccination rates, or which experienced the COVID-19 pandemic starting at widely divergent time points). This study, therefore, uses a curated dataset of 51 nations that experienced the beginning of the COVID-19 pandemic at comparable times, have high rates of SARS-CoV-2 testing so that incidence of COVID-19 can be adequately tracked, and for which complete data on vaccination rates are available. These data permit the impacts of various vaccinations to be compared more appropriately than has previously been possible. Previous studies have also almost universally failed to explore possible interactions among vaccinations in protecting against COVID-19 or to control for confounding effects of multiple vaccinations being given in time-clustered sets. This study performs both types of analysis.

This study goes on to test further implications of the disease burden hypothesis regarding the relationship of COVID-19 disease and death rates with regard to population rates of lower respiratory infection (LRI) and invasive pneumococcal disease (IPD). LRI should correlate with COVID-19 rates if viral infections such as influenzae, parainfluenzae, adenoviruses, and respiratory syncytial virus, or bacterial infections such as *Chlamydia pneumoniae, Mycoplasma pneumoniae,* and *Staphylococcus aureus*, are predisposing factors for SARS-CoV-2 acquisition. IPD should correlate with COVID-19 rates better than LRI if pneumococcal infections (e.g., *Streptococcus pneumoniae, Streptococcus pyogenes*) play a particular role in predisposing to COVID-19 disease.

Finally, the various observations that are derived from the curated set of 51 nations are tested more specifically in local regions of Italy and the United States. National-level datasets can and do often hide variations that are more revealing at more granular levels of analysis. In this case, data for the regions of Italy and the states of the U.S. permit a more detailed analysis of whether national-level trends in vaccinations with influenza, MMR, and pneumococci correlate equally well at local levels as well. Correlations between LRI and IPD rates with COVID-19 rates are similarly tested at these local levels.

In sum, this paper unfolds as follows. Current literature concerning the relationship of coinfections with COVID-19 is summarized and the possible impact of vaccinations in moderating COVID-19 risks reviewed. These points set up the hypothesis to be tested, which is that disease burden determines COVID-19 risks. An international dataset is then used to test this hypothesis, first by analyzing the relationship of vaccination rates and COVID-19 case and death rates, and second by analyzing the relationship of LRI and IPD to COVID-19 case and death rates. Results are controlled for possible interactions among vaccinations and by examining whether the international data also hold up at regional and state levels within Italy and the United States. Conclusions are provided, and limitations of the study are discussed.

### 1.1. Coinfections in Severe COVID-19

The specific hypothesis to be tested in this paper is that non-SARS-CoV-2 respiratory infections significantly increase the probability of severe COVID-19 and therefore that vaccination against these microbes results in significant protection against symptomatic and, in particular, severe COVID-19 [5,6]. This section summarizes the current literature concerning the existence of such coinfections among COVID-19 patients and the implications of these coinfections for the severity of the disease.

Respiratory viruses are well-known to condition the lungs to be more susceptible to bacterial super-infection [7] and were highly associated with severe disease and death in previous respiratory virus pandemics such as the Great Influenza pandemic of 1918–1919 [8,9,10,11] and severe acute respiratory disease (SARS) (reviewed in [12]). In particular, pneumococcal infections are found in up to 25% of people over the age of 60 with influenza-like illnesses regardless of viral cause [13]. Thus, while the pneumonias associated with SARS-CoV-2 were initially attributed exclusively to the virus, recent investigations have revealed high levels of fungal and bacterial coinfections in COVID-19 cases comparable to or higher than those found in hospitalized influenza cases [12,13,14,15,16,17]. Lai et al. [16], for example, have reviewed some of the relevant literature, concluding that “The prevalence of coinfection was variable among COVID-19 patients in different studies, however, it could be up to 50% among non-survivors. Co-pathogens included bacteria, such as Streptococcus pneumoniae, Staphylococcus aureus, Klebsiella pneumoniae, Mycoplasma pneumoniae, Chlamydia pneumonia, Legionella pneumophila and Acinetobacter baumannii; Candida species and Aspergillus flavus; and viruses such as influenza, coronavirus, rhinovirus/enterovirus, parainfluenza, metapneumovirus, influenza B virus, and human immunodeficiency virus. Influenza A was one of the most common co-infective viruses.” Chen et al. [12] also concluded that bacterial and fungal coinfections were common among hospitalized COVID-19, “increasing disease severity”.

More specifically, in Spain, Cucchiari et al. [18] reported five COVID-19 cases coinfected with *Streptococcus pneumoniae,* which prompted Garcia-Vidal et al. [19] to test 949 consecutive COVID-19 patients for bacterial lung coinfections, of which 74 were given detailed laboratory workups at diagnosis: 40.5% were found to have a bacterial infection, of which 57% were *S. pneumoniae*, 29% *Staphylococcus aureus*, and 10% *Haemophilus influenzae* (some patients had multiple infections). A French study of 92 severe adult COVID-19 patients by Contou et al. [20] found that 28% had bacterial coinfections upon hospital admission consisting mainly of *S. aureus* (31%), *H. influenzae* (22%), *S. pneumoniae* (19%), and *Enterobacteriaceae* (16%). In an Italian study, 13% of 575 COVID-19 pneumonia patients tested positive in a urine antigen test for *S. pneumoniae,* but other bacteria were not investigated (Desai et al., 2020). In China, Zhu et al. [21] performed a look-back study at 257 laboratory-confirmed COVID-19 cases using polymerase chain reaction methodology to identify the presence of coinfections in lung autopsies. They found that 94% had one or more bacterial coinfections with S. pneumoniae (about 60%), followed by *Klebsiella pneumoniae* (55%), *H. influenzae* (40%), *S. aureus* or *S. haemolyticus* (20%), and Escherichia coli (15%). Again, many patients had multiple bacterial infections. These infections were found mainly in patients aged 16–44, with slightly lower rates in patients 44–64 and with the lowest rates in the youngest and oldest patients. Lv et al. [17] found similar rates of infection with Staphylococci and E. coli as did Zhu et al., but did not test for Streptococci, Klebsiella, and Haemophilus species. Zhou et al. [22] reported that 56% of COVID-19 patients were coinfected with at least one pathogenic bacterium (21% with multiple bacteria), of which Streptococcus, Haemophilus, and Pseudomonas species were the most common. Wang et al. [23] found that 53% of elderly patients diagnosed with COVID-19 in Wuhan, China had bacterial coinfections; these were rarer among those who survived (34% coinfected) as compared with those who died (82% coinfected). In another Chinese study, Zhang et al. [24] found that severe cases of COVID-19 were characterized by the presence of bacterial coinfections (25%), which were rare in mild cases (3%), and 55% of the patients who died had such coinfections. Unfortunately, the bacterial infections were not identified in either the Wang et al. [23] or Zhang et al. studies [24]. However, it may be assumed that these were similar to those described by Zhu et al. [21] above.

Other studies have found lower rates of bacterial coinfections but often for methodological reasons. Many studies reporting low rates of bacterial infections involved patients who had already begun multi-regimen antibacterial therapies that would have masked most infections. For example, among 476 COVID-19 patients, of which 93% were receiving antibacterial therapy, Feng et al. [25] found that 34.5% still had detectable bacterial coinfections. Similarly, Zhou et al. [26] observed that 15% of their 191 COVID-19 patients had bacterial coinfections even while 95% of the cohort were being treated with antibacterial therapy; 27 of 28 of those with bacterial coinfections died. A similar study in the U.S. by Townsend et al. [27] put 84 of 117 COVID-19 patients on intravenous antimicrobial therapy for empirically diagnosed lower respiratory infection, and 11 for other sources of bacterial infection such as genitourinary; of these, 40 required ICU admission and 6% subsequently tested positive for bacterial coinfection with E. coli, Staphylococcus or Streptococcus bacteria. Rothe et al. [28] similarly found low rates of bacterial coinfections in German COVID-19 patients, but over 75% were on multiple antibiotic regimens, and no urinary antigen tests were performed or respiratory samples obtained from any patient. Other methodological difficulties have involved the absence of appropriate pathology specimens. For example, in a U.K. look-back study, Hughes et al. [29] lacked appropriate urine, sputum, and respiratory samples for between 70 and 93% of their 836 patients and reported bacterial blood infections in only 7.1%, of which they wrote off half as contaminants. Similarly, Ma et al. [30] tested for the presence of bacterial antibodies in COVID-19 patient serum, finding that about 5% had either Chlamydia pneumoniae (5.2%) or M. pneumoniae (4.4%), but they did not test for Streptococcal, Klebsiella, or Staphylococcal antibodies. The most important methodological failing is that most clinicians do not test for bacterial infections in COVID-19 either because they have been told that SARS-CoV-2 accounts for all respiratory symptoms or because SARS-CoV-2 masks the presence of bacterial coinfections [16,31]. Some meta-studies have ignored these methodological limitations, concluding that bacterial coinfections are rare in COVID-19 (e.g., [32,33,34]). Increasingly, however, clinicians are becoming aware of the threat that bacterial coinfection poses in COVID-19. Bengoechea and Bamford [35], Lai et al. [16], and Huttner et al. [36] all conclude, in the words of Vaillancourt and Jorth [31], that there is “an unrecognized threat of secondary bacterial infections with COVID-19”.

Several viruses have also been found to co-infect COVID-19 patients. Three percent of COVID-19 patients have been found to have either influenza A or respiratory syncytial virus coinfections, and parainfluenza or metapneumovirus infections also occur [12,16,22,34,37,38]. Such coinfections are not, however, associated with more severe morbidity and higher mortality than uncomplicated SARS-CoV-2 infections [30,38,39,40,41].

### 1.2. Non-SARS-CoV-2 Vaccinations and COVID-19 Risk

Given the types of coinfections found in moderate-to-severe COVID-19 patients, vaccination against *S. pneunomiae* and *H. influenzae* might be expected to protect some patients against severe COVID-19 following SARS-CoV-2 exposure and, indeed, two types of studies have found that such vaccinations may be protective. National rates of vaccination with pneumococcal vaccines correlate inversely with case and death rates from COVID-19 [5,42] and pneumococcal vaccination has been demonstrated to lower the risk of coronavirus infections associated with pneumonias in children and adolescents [43]. Additionally, rates of pneumococcal vaccinations correlate inversely with the risk of COVID-19 in four separate cohort or case-controlled studies involving tens to hundreds of thousands of patients each [44,45,46,47]. More specifically, Sambul et al. [48] found that antibody titers to pneumococcal vaccine antigens were correlated inversely with the risk of COVID-19: non-infected individuals averaged titers that were four times higher than those of COVID-19-infected individuals and (*p* = 0.002) and people with asymptomatic infections had significantly higher titers than people with severe cases (*p* = 0.01) [48]. Notably, this study also demonstrated that the risk of COVID-19 and its severity were not associated with immunodepression since titers of measles, tetanus, and *Haemophilus influenzae* type B (Hib) antibodies were higher among COVID-19 patients than among healthy controls, and the two groups displayed equal antibodies titers against mumps, diphtheria, and varicella-zoster virus [48].

While Pawlowski et al. [45] found in their case-controlled study that the strongest protection against COVID-19 was produced by pneumococcal vaccination, they also reported that *Hemophilus influenzae* type B (Hib), measles-mumps-rubella (MMR), varicella-zoster virus, geriatric flu, and hepatitis A/hepatitis B (HepA-HepB) vaccines administered within five years of COVID-19 testing predicted a negative SARS-CoV-2 test. However, as they note [45], some of the vaccinations occurred within similar time frames for the patient, so confounding effects were difficult to tease apart. Nonetheless, other population-level studies have also identified COVID-19 protective effects of influenza vaccination [49,50,51,52,53,54,55] and MMR vaccination [56,57,58], or more specifically, rubella vaccination [48]. Again, except for the Sambul et al. [48] study, data were not controlled for possible confounding effects of receiving multiple vaccines within a similar timeframe, but in several controlled studies, neither MMR (or measles-containing and rubella-containing vaccines) nor influenza vaccine was associated with protection against COVID-19 case or death rates in population-wide studies [5,59,60]. Moreover, even in Sambul et al.’s [48] study, in which they found a significant correlation between higher rubella titers and lower risk of COVID-19, further analysis revealed no relationship (*p* = 0.89) between rubella antibody titers and disease severity. Case-controlled studies of influenza vaccination have reached contradictory conclusions about its effectiveness as a COVID-19 prophylactic: Martínez-Baz et al. [61] and Belingheri et al. [62] found no benefit; Ragni et al. [63] found a benefit only for patients over the age of 65 but not among younger vaccine recipients; Noale et al. [46] found exactly the reverse, with the only benefit among younger recipients; while Massoudi et al. [64] reported significant protection in all age groups.

The anti-tuberculosis vaccine, Bacillus Calmette–Guerin (BCG), has also been investigated for potential protection against COVID-19, with a number of population studies suggesting its effectiveness in preventing COVID-19 cases and deaths (e.g., [59,60]), but the most rigorous and best-controlled population studies have not found a correlation between BCG vaccination rates and lower COVID-19 case or death rates (extensively reviewed in [65,66,67,68]), either at a national level [5,44,45,46,68] or in well-controlled cohort studies in Israel, Germany, and Sweden [69,70,71]. However, a study of U.S. healthcare workers (among whom BCG vaccination is very rare) found significant lower COVID-19 antibody titers (though, note, not confirmed SARS-CoV-2 infections!) among those who had been BCG vaccinated than those who had not, while no differences in SARS-CoV-2 titers were found by influenza A, pneumococcal or meningitis vaccination [72]. A case-controlled study in the Netherlands found that recent BCG vaccination (within 5 years) was associated with a small decrease in the odds of a COVID-19 diagnosis and slightly decreased symptomology [73]. Thus, while the preponderance of evidence argues against COVID-19 protective effects of either BCG or MMR vaccination, some weaker studies support their possible efficacy.

Finally, diphtheria-tetanus-pertussis vaccination [74,75,76,77] and polio vaccination [78] have also been suggested as possible prophylactics against COVID-19, but at present, no ecological study exists to support these possibilities. Published studies have found no protective benefit [5,44,45], and antibody titers against tetanus do not correlate with COVID-19 protection [48]. However, Sambul et al. [48] reported a very significant inverse correlation between pertussis antibody titers and risk of COVID-19 (*p* = 0.0001) in a case-controlled study, but these titers did not distinguish between asymptomatic and severe cases (*p* = 0.49) or mild and severe cases (*p* = 0.11) of COVID-19 so that while pertussis vaccination may protect against SARS-CoV-2 infection, it does not seem to moderate disease severity. Notably, the role of pertussis itself as a super-infection in COVID-19 is also contradictory: pertussis superinfections of SARS-CoV-2 were significantly increased over control cases of other viral pneumonias in one Chinese study [79], but a Brazilian study found no *B. pertussis* among their COVID-19 patients [80]. Thus, the role of pertussis in COVID-19 and the efficacy of pertussis vaccination are in need of further investigation.

In sum, while many vaccines have been proposed as prophylactics against COVID-19, few, other than pneumococcal vaccines, have produced unambiguous evidence of protection in both ecological or population-wide, controlled cohort and/or antibody titer studies. Other vaccines that may afford protection include influenza, rubella, and pertussis. The preponderance of current evidence points to there being no protection afforded by Hib, BCG, diphtheria, tetanus, measles, mumps, or polio vaccines.

### 1.3. Purpose of The Current Study

The purpose of the current study is to extend a previous ecological study of the effects of vaccines on COVID-19 case and death rates [5] by increasing the number of countries from 24 to 51 for which vaccination rate data exist not only for childhood vaccinations but also vaccinations in adults 65 years of age and older with pneumococcal and influenza vaccines and, where available, for pneumococcal vaccinations in high-risk adults ages 18 to 65. The current study is unique in also testing for possible additive and synergistic vaccination effects in terms of COVID-19 prophylaxis and for controlling for co-variance among vaccines. Since many vaccines are given concurrently or in close time proximity to others, rates of some vaccinations may correlate with others confounding the determination of which vaccines are actually effective as prophylactics for COVID-19. In addition, a novel approach to testing the potential role of pneumococci and other respiratory microbial infections in COVID-19 is implemented by comparing COVID-19 cases and death rates to national (and in some cases regional) rates of invasive pneumococcal disease (IPD) and lower respiratory infections (LRI).

The hypothesis to be tested is that if pneumococcal vaccination is truly effective as a protection against the risk of COVID-19 and specific in its effects, then COVID-19 case and/or death rates should correlate with IPD rates but not LRI, the latter being a much broader disease category encompassing a far wider range of microbial infections. The other vaccines mentioned above (DTP, polio, BCG, MMR, and influenza) should have no significant correlation with invasive pneumococcal disease (IPD) rates. Pneumococcal vaccination, however, should have a minimal impact on lower respiratory infection (LRI) rates since LRI have a very wide range of causes, including, notably, influenza; thus, influenza vaccine should decrease the incidence of LRI. Finally, it may be possible that some of the vaccines have additive or synergistic effects that increase their effectiveness in lowering the risk of COVID-19 disease and/or death; an example might be a combination of pneumococcal and Hib or pneumococcal and influenza vaccines.

## 2. Methods

### 2.1. International Data

In order to test the hypothesis, data were obtained regarding vaccination rates for the most recent years available (usually between 2016 and 2019) for pneumococcal vaccines among children, at-risk adults between the ages of 18 and 65, and adults over the age of 65; Hib vaccination rates; adult influenza vaccination rates; diphtheria-tetanus-pertussis vaccination rates; measle-containing virus vaccination rates; oral polio vaccination rates; Bacillus Calmette–Guerin (tuberculosis) vaccination rates; COVID-19 cases per million people as of 24 November 2020; COVID-19 deaths per million people as of 24 November 2020; lower respiratory infection (LRI) deaths per thousand people in 2019; and the most recent invasive pneumococcal disease (IPD) cases per one hundred thousand people (usually between 2016 and 2019). Fifty-one countries were included in the current study, chosen by several criteria: (1) high rates or SARS-CoV-2 testing (at least 400,000 per million residents); (2) availability of national rates of obesity and diabetes and percent of population 65 years of age or older; and (3) the availability of data corresponding to all of the vaccination and disease rate categories just listed, with three exceptions: a country was still included in the study even if IPD data, and/or pneumococcal vaccination rates among at-risk adults and/or adult influenza vaccination rates could not be located.

Data were accumulated from a previous study of 24 of the countries [5] and, where possible, from several general sources such as the Worldometer COVID-19 website (https://www.worldometers.info/coronavirus/coronavirus-cases/#daily-cases, accessed on: 24 November 2020), the World Health Organization Immunization Monitoring website (https://apps.who.int/immunization_monitoring/globalsummary/timeseries/tscoveragebcg.html, accessed on: 29 November 2020), and national rates of lower respiratory infections (LRI) from [81]. Additional data were accumulated from reviews and individual studies identified through PubMed (National Library of Medicine, Bethesda, MD. USA) and Google (Alphabet, Inc., Mountain View, CA, USA) searches, including relevant keywords. In general, sources were limited to peer-reviewed studies or official websites of public health agencies of the various nations and, with a few exceptions, were published in English, French, or German. In cases where sources provided different figures (as occurred when studies of vaccination rates were performed in different locations within a single country), all available data were accumulated, and the average of the data used for further analysis.

Additionally, data regarding pneumococcal vaccination rates were modified to reflect the type of pneumococcal conjugate vaccine (PCV) used in each nation: the data were used without modification if PCV13 (which covers the thirteen most common pneumococcal strains) was used; data were multiplied by 10/13 if PCV10 (which covers ten pneumococcal strains) was used; and data were multiplied by 7/13 if PCV7 (which covers only seven pneumococcal strains) was employed. Further, since pneumococcal vaccinations are carried out in infancy, adulthood, and old age, the rates of vaccination at each age group were added to obtain their sum, which acted as an approximation of the overall protection in a population. Where data were available for infancy, at-risk individuals between the ages of 18 and 65, and people 65 years of and over, all three sets of data were added. One can imagine these data as being a number that could approach 300% if all groups were completely vaccinated. In practice, the sum rarely approached 200. If, however, no data could be found regarding vaccination in at-risk groups, then the infant and old-age data were summed, and the total possible was 200%. Data regarding pneumococcal vaccination rates for these summation groups were analyzed separately for correlations to the other factors in the study.

Invasive pneumococcal disease (IPD) data were accumulated from multiple sources (which occasionally also contained pneumococcal vaccination data): Europe [82,83,84,85]; Eastern Europe [86]; German [87]; Italy [88,89,90]; Israel [91]; Egypt [92]; Turkey [93]; Korea [94]; Japan [95]; Australia [96]; New Zealand [97]; U.K. [98]); U.S. [99,100]); Argentina [101]; Brazil ([102]; Canada [103]; Mexico [104]; Peru [105]; Chile [106]; South Africa [107].

Adult and at-risk pneumococcal and influenza vaccination data were obtained from [5] and: Andorra [108]; Argentina [109,110]; Belgium [111]; Brazil [112]; Central and Eastern Europe [113]; Denmark [114,115]; European nations [85,116,117,118,119]; Europe and Gulf nations [120]; Germany [121]; Gulf nations [122,123]; Hong Kong [124]; Hungary [125]; Iceland [126]; Ireland [127]; Israel [128]; Latin America [129]; Mexico [130,131]; New Zealand [132]; Poland [133]; Qatar [134]; San Marino [135]; Singapore [136]; Slovenia [137]; South Africa [138]; Sweden [139]; Switzerland [140,141,142]; Taiwan [143,144,145]; Turkey [146,147,148]; and U.K. [149]. All of these data are summarized in Table 1.

Percent of population over the age of 64 was obtained from the World Bank, https://data.worldbank.org/indicator/SP.POP.65UP.TO.ZS (accessed on: 3 March 2021); national rates of obesity were obtained from the OECD Obesity Update 2017. https://www.oecd.org/health/obesity-update.htm (accessed on: 3 March 2021); while diabetes prevalence (% of population ages 20 to 79) Country Ranking, 2020. https://www.indexmundi.com/facts/indicators/SH.STA.DIAB.ZS/rankings (accessed on: 3 March 2021). Some of these data can be found in [5] but are not included in Table 1 in order to keep the table size within reasonable limits.

### 2.2. Italian Regional Data

A second dataset was accumulated to compare the importance of childhood pneumococcal [88], measles-mumps-rubella [150], and adult influenza [49] vaccination rates on rates of invasive pneumococcal disease [87,88,151] and COVID-19 death rates [152] in the twenty-one regions of Italy. Rates of adult pneumococcal and other vaccinations such as polio, diphtheria, and BCG were not found for more than a handful of regions despite a dedicated search of both PubMed and Google and were insufficient to support statistical analysis.

### 2.3. U.S. Data

Finally, a third dataset was accumulated to compare by state, within the United States, COVID-19 deaths per 1 million population; cases per 1 million population; percent of children under the age of five vaccinated with the PCV-13 pneumococcal vaccine [153]; percent of adults 18–64 years of age vaccinated with a pneumococcal vaccine [154]; and adults 65 years of age and older who had received a pneumococcal vaccination [155]. Case and death rates were obtained from Worldometer (https://www.worldometers.info/coronavirus/coronavirus-cases/#daily-cases, accessed on: 9 November 2021) on 9 November 2020. Influenza vaccination rates among adults 65 years and older by state were accessed for the 2019–2020 season from https://www.americashealthrankings.org/explore/senior/measure/flu_vaccine_sr/state/ALL (accessed on: 6 March 2021), and death rates per 100,000 population from pneumonias (lower respiratory infections) was acquired from the National Center for Health Statistics, https://www.cdc.gov/nchs/pressroom/sosmap/flu_pneumonia_mortality/flu_pneumonia.htm (accessed on: 6 March 2021). Additional information for pneumococcal vaccination rates for Cook County and Chicago, Illinois, was obtained from a Chicago Tribune website (https://www.chicagotribune.com/coronavirus/ct-viz-covid-19-cases-by-zip-code-20200407-aikakoyycje4fbqvferzjffkg4-htmlstory.html, accessed on: 9 November 2021), also accessed on 9 November 2020. These data were accumulated in Table 2. As with the regions of Italy, a dedicated PubMed and Google search was unable to locate rates of other vaccinations relevant to this study for sufficient states to make statistical analysis possible.

### 2.4. Statistics

Correlation coefficients (R^2^) and Pearson’s correlation coefficient (R) between the various factors aggregated in Table 1 and Table 3 were calculated using Agrimetsoft (https://agrimetsoft.com/calculators/correlation%20coefficient, accessed on: 17 April 2021). R provides a measure of how significant the correlation between any two factors is, on a scale from 0 to +/-1, with 0 being no correlation and 1 being complete positive correlation and -1 being a completely negative (or inverse) correlation. In general, an absolute value of R below 0.3 is considered to be a weak correlation; between 0.3 and 0.5 a moderate correlation; and above 0.5 a strong correlation. R^2^ provides a measure of “effect size”, or how much of the effect on an outcome is due to the particular factor to which it is correlated, which can vary from 0 to 1 and can only have a positive value.

Since multiple correlations were run on each dataset, a Bonferroni correction was applied to each set of correlations. Curtin and Schulz [156] provide a convenient graph for estimating such corrections. These estimates were confirmed by a second method, which was to convert the R values into probability values, *p*, using a *p*-value for Correlation Coefficients Calculator (https://www.danielsoper.com/statcalc/calculator.aspx?id=44, accessed on: 27 April 2021) to determine the R value required to achieve *p* < 0.05. A direct correction program (F-Value and *p*-Value Calculator for Multiple Regression) was used to determine the R^2^ value required to achieve *p* < 0.05 given multiple regression factors (https://www.danielsoper.com/statcalc/calculator.aspx?id=15, accessed on: 27 April 2021). The R and R^2^ values required to achieve *p* < 0.05 vary by dataset and are provided in the caption associated with each Table in the Results section.

## 3. Results

### 3.1. Data Sets

The data sets accumulated through the processes outlined in the Methods section are found in Table 1, Table 2 and Table 3. Table 1 summarizes the data for COVID-19 case and death rates, lower respiratory (LRI) and invasive pneumococcal disease (IPD) rates, and vaccination rates for 51 nations. Table 3 summarizes data for the regions of Italy regarding the rates of COVID-19 deaths per 100,000 (100K) population; invasive pneumococcal disease (IPD) per 100,000 (100K) population; percent of infants vaccinated with the PCV13 pneumococcal vaccine (PCV13); percent of infants vaccinated with measles-mumps-rubella vaccine (MMR); percent of 16 year-olds vaccinated with MMR; and the percent of the population vaccinated against influenza (INF). T Data concerning the rates of COVID-19 cases and deaths in the 50 states of the United States of America; the rates of pneumococcal vaccination (PNEUM) among children, at-risk adults 18 to 64 years of age, and adults 65 years of age and older; the sum of those pneumococcal vaccination rates; and influenza vaccination (INF) rates among adults 65 years of age and older and among the general population of all ages. Three major cities are also included: Chicago, IL (Illinois); Philadelphia, PA (Pennsylvania) and Washington, District of Columbia.

### 3.2. Correlations between National Vaccination Rates and COVID-19 Case and Death Rates

The first step in testing the hypothesis that pneumococcal, but not other, vaccinations protect against COVID-19 acquisition and/or severity was to correlate COVID-19 case and death rates with pneumococcal and other vaccination rates. Data from the 51 nations listed in Table 1 were used to perform these correlations, which are summarized in Figure 1.

Data concerning national COVID-19 case and death rates (per million population), with national vaccination rates for pneumococcal vaccines during infancy, for at-risk adults and for adults 65 years of age and older; influenza vaccination rates in adults 65 years and older; Bacillus Calmette–Guerin (BCG)—or tuberculosis—vaccination rates; oral polio vaccination rates; measles-mumps-rubella (MMR) vaccination rates; diphtheria-pertussis-tetanus (DPT) vaccination rates; and Haemophilus influenzae type B (Hib) vaccination rates; and the results are displayed in Figure 1. Strong correlations were found between higher rates of childhood pneumococcal vaccination and lower rates of COVID-19 death, which were amplified by adding pneumococcal vaccination rates among adults 65 years of age and older and/or pneumococcal vaccination rates for at-risk adults (those with underlying, predisposing diseases) between the ages of 18 and 65. The R values, or effect sizes, suggest that combined pneumococcal vaccination across all age groups may account for as much as half of the differences among COVID-19 death rates observed between different nations. Notably, the negative correlation between pneumococcal vaccination and COVID-19 death rates is as strong as the correlation between COVID-19 death rates and COVID-19 case rates, which provides a useful benchmark for evaluating the pneumococcal vaccine/COVID-19 relationship.

Few additional significant results were found among the correlations listed in Figure 1. Childhood pneumococcal vaccination rates and combined pneumococcal vaccination rates had much stronger inverse correlations with both COVID-19 death and case rates than any other vaccine explored in this study, but these were only moderate correlations. The only other vaccine displaying any significant protective effect was MMR, which exhibited a weakly significant, negative correlation with COVID-19 death rates. Correlations with COVID-19 case rates were weaker for all vaccines than with COVID-19 death rates and retained significance only for infant pneumococcal vaccination rates and some combinations of the infant pneumococcal rate with adult pneumococcal rates. Thus, while pneumococcal vaccination appears to protect strongly against COVID-19 deaths, it has only a moderate effect on the risk of SARS-CoV-2 infection. No other vaccine appears to confer any significant protection against COVID-19 infection.

### 3.3. Do Combinations of Vaccines Protect against COVID-19?

Since people almost invariably are inoculated with multiple vaccines during their lifetimes, it is important to control for the possibility that some vaccines may add to or synergize with pneumococcal vaccines to confer additional protection against COVID-19 disease or death. Thus, the data in Table 1 were aggregated in various permutations of the vaccination rate data and explored further to assess whether combinations of vaccines might provide better protection than individual vaccines. Figure 2 presents the results of the combinations of pneumococcal vaccination rates with the rates of the other vaccines used in the study. For simplicity, only the results from the total combined pneumococcal vaccination rates from Figure 1 were used in this further analysis. It is also important to note that Figure 2 does not display any of the rate data comparisons from the other possible pairs of vaccines in the study, all of which were calculated, but none of which showed increases that boosted them into a statistically significant range (R > 0.350 for a *p*-value of <0.05).

The results displayed in Figure 2 show that adding pneumococcal vaccination rates to rates of vaccination with other vaccines produces no additional protective effect and, in fact, tends to decrease the significance of the resulting correlation. In other words, the result of looking at the sum of two vaccination rates is to average their individual effects, thereby lowering the observed correlation with COVID-19. One important exception, however, stands out, and that is adding the rate of influenza vaccination in adults 65 and older, which is in-and-of-itself not- significantly correlated with COVID-19 protection, to the total pneumococcal vaccination rate provides significantly higher protection against the risk of COVID-19 infection (as measured by the COVID-19 case rate). Thus, influenza vaccination rates are additive with pneumococcal vaccination rates resulting in apparently increased protection against COVID-19. Indeed, the same increase was found in adding the influenza rate to the infant-plus-65 and older pneumococcal rate, to the infant pneumococcal rate, or to the 65-and-over pneumococcal rate (data not shown). Thus, there appears to be a possible protective synergism between pneumococcal and influenza vaccines in terms of preventing the acquisition of SARS-CoV-2. Equally notably, although MMR vaccination has a moderately protective effect on COVID-19 rates of death in Figure 1, MMR vaccination provided no additional protection when added to pneumococcal vaccination rates (Figure 2).

### 3.4. Controlling for Possible Confounding of Vaccine Effects

The results of Section 3.2 and Section 3.3 indicate that the total rate of pneumococcal vaccination across a lifetime is correlated with decreased COVID-19 case and death rates (Figure 1); that influenza vaccination among the elderly may improve pneumococcal vaccine protection against COVID-19 acquisition (Figure 2); and that MMR vaccination may produce weak, independent protection against COVID-19 death (Figure 1). Because individuals often receive more than one vaccine at any particular age, and (as can be seen by inspection of Table 1) nations that vaccinate broadly against one set of diseases often vaccinate broadly against others, further controls were run to determine the risk that the correlations observed in Figure 1 and Figure 2 were due to confounding factors such as people 65 and older getting both a pneumococcal and an influenza vaccine. Similarly, it is possible that infants getting pneumococcal vaccination obtained other childhood vaccines at similar rates so that some part of the apparent protection afforded by pneumococcal vaccination is due to confounding effects of the co-administered vaccines. Notably, such controls have only been implemented in one previous study [45], making these calculations of significance beyond the present study.

Figure 3 displays the probabilities that people in any given nation will receive any pair of the vaccines listed in Table 1. Notably, there is no significant correlation between rates of pneumococcal vaccination and rates of influenza vaccination in most nations, suggesting that these can be treated as independent variables in considering the results in Figure 2. This result confirms the additivity of pneumococcal and influenza vaccination in lowering the risk of COVID-19 acquisition.

Rates of Hib, DPT, and MMR, however, correlate very strongly so that the effects attributed to one may actually reflect an effect produced by one or more of the others working together. Additionally, rates of childhood pneumococcal vaccination correlate weakly-to-moderately with rates of MMR, Hib, and DPT vaccination so that a possible contribution of these vaccines to the observed protective effect of pneumococcal vaccination cannot be ruled out completely. The main arguments against such a contribution are the lack of any significant negative correlation between MMR, Hib, or DPT vaccinations and COVID-19 case or death rates in Figure 1 and the lack of observed additivity between pneumococcal vaccination rates and the rates of any of these other vaccines (Figure 2).

### 3.5. Pneumococcal Invasive Disease and Lower Respiratory Infections as Possible Risks for COVID-19

Thus far, the results suggest that pneumococcal vaccination varies inversely with COVID-19 case and, especially, death rates as predicted by this paper’s hypothesis. Influenza and MMR vaccines may play a minor and independent role in such protection and may act additively with pneumococcal vaccination. An additional test of the hypothesis that pneumococcal vaccination, in particular, protects against COVID-19 is, therefore, to look for a correlation between invasive pneumococcal disease (IPD) and COVID-19. Logically, if pneumococci are important co- or superinfections in the pathogenesis of COVID-19, then IPD rates should correlate with the risk of COVID-19 acquisition and particularly with severe disease leading to death. In other words, where IPD rates are high, COVID-19 rates should be high. Additionally, IPD rates should be high where pneumococcal vaccination is low.

However, it is also possible that pneumococci play no special role in COVID-19 acquisition or disease development and that other lung infections also synergize with SARS-CoV-2 to enhance the risk of disease and death. One way to test this possibility is to examine whether there is a correlation between the rate of lower respiratory infections (LRI) and COVID-19 rates of infection and death. If LRI rates correlate with COVID-19 rates as well as or better than do IPD rates, then pneumococci are only one of many possible microbes that might be enhancing COVID-19 risks. On the other hand, if IPD correlates with COVID-19 rates and LRI does not, then this result would indicate a special role of pneumococci in COVID-19 acquisition and/or disease development.

These possibilities were explored by examining the relationships between COVID-19 case and death rates, rates of LRI and IPD, and how vaccination rates correlate with these (Figure 4).

In sum, IPD but not LRI rates predict rates of COVID-19 deaths and, more weakly, rates of COVID-19 cases as well, making pneumococcal disease a specific risk in COVID-19.

Figure 4 demonstrates that pneumococcal vaccination is generally protective against invasive pneumococcal disease (IPD) but is not protective against lower respiratory infections more generally. There is no significant correlation between IPD risk and LRI risk for any population age group suggesting that rates of pneumococcal disease are largely independent of rates of influenza and other common causes of respiratory disease and that, for the present purposes, IPD and LRI rates can be treated as statistically independent variables. LRI rates correlated significantly and inversely only with the rates of vaccination with influenza (among adults 65 and older), which makes sense since influenza is one of the main causes of LRI. Additionally, there is only a weak correlation between COVID-19 rates of death and national rates of LRI and no significant correlations between LRI rates and COVID-19 case rates. Thus, the general risk of respiratory infection (LRI) among a population is not a suitable predictor of COVID-19 case rates and only a weak (and not statistically significant) predictor of COVID-19 rates of death.

On the other hand, COVID-19 rates of death are moderately to strongly correlated with rates of IPD among the general population of nations and both for the general population and among the elderly. COVID-19 case rates are less well correlated, but still statistically significantly so, with IPD rates except among the elderly. In addition, as predicted, pneumococcal vaccination rates are moderately to strongly inversely correlated with IPD rates, especially among the elderly, indicating that pneumococcal vaccination confers a reasonable degree of protection against IPD within a population. No other vaccine rate (MMR, influenza, DTP, polio, or BCG) correlated significantly with any measure of IPD or LRI (data not shown).

### 3.6. Testing the Hypothesis in the Regions of Italy

A further test of the hypothesis that vaccination against pneumococci provides protection against COVID-19 and that IPD rates should directly predict the risk of COVID-19 cases and death rates is to determine whether these national trends are also observed within regions of various nations. One example is provided by Italy (Figure 5). IPD rates in regions of Italy correlated very significantly with rates of COVID-19 deaths (Figure 5 and Figure 6). Unfortunately, regional data for pneumococcal vaccination rates were not found for at-risk adults or adults 65 years of age or older. The only data available were for infants receiving PCV13, which was only weakly (and not significantly) protective against IPD and not significantly protective against COVID-19 death. However, a significant inverse correlation between rates of COVID-19 deaths in regions of Italy and rates of influenza vaccination was observed, as predicted from the synergy of influenza and pneumococcal vaccination suggested in the international data summarized in Figure 2. The direct correlation between IPD and COVID-19 death rates, as well as the inverse relationship between influenza vaccination rates and COVID-19 death rates and IPD rates, is also clearly evident in the maps of Italy compared in Figure 6. Notably, the weak negative correlation between MMR vaccination rates and risk of COVID-19 disease found at the international level is not replicated within Italy, where there MMR vaccination rates have no predictive value with regard to COVID-19.

### 3.7. Testing the Hypothesis in the United States

Data from the United States of America provides an additional state-wide test of the generalizability of the correlations observed at an international level. Once again, within the United States, a strong inverse correlation (R = −0.577, R^2^ = 0.333) exists between overall rates of pneumococcal vaccination and rates of COVID-19 deaths in individual states and major metropolitan areas such as the District of Columbia and the greater Chicago and New York City (Figure 7 from Table 2). As with the international data, there is a significantly weaker inverse correlation between COVID-19 case rates and pneumococcal vaccination rates that, in this case, fails to achieve statistical significance (Figure 7). Influenza vaccination rates did correlate significantly with COVID-19 case or death rates, as was also seen in the international (Figure 1). Death rates from lower respiratory infections (LRI) once again failed to correlate with COVID-19 case or death rates, as was previously found internationally (Figure 1). Unfortunately, state-by-state data on invasive pneumococcal disease (IPD) rates could not be located, so it was not possible to determine whether IPD rates predicted COVID-19 rates as was found internationally (Figure 4) and in Italy (Figure 6).

However, other U.S. data do provide evidence that IPD rates do correlate with COVID-19 rates and are, in fact, better predictors than rates of pneumococcal vaccination for at least some risk groups. As has been pointed out before with regard to COVID-19 risks [5,45], population averages certainly hide ethnic disparities in access to and use of healthcare resources such as vaccinations. In particular, minority populations in the U.S., U.K., Canada, and other nations often contract and die of COVID-19 at higher rates than do majority populations [157,158,159]. Within the United States, this disparity is very evident in Table 4, which illustrates the correlation between reduced pneumococcal vaccination rates and corresponding increases in invasive pneumococcal disease among Native Americans, African Americans/Blacks, and Hispanic/Latinos as compared with Whites and the correspondingly higher rates of COVID-19 hospitalizations and deaths as has been found in the international data (Figure 1) Note that the hospitalization and death rates among most minorities, other than Asians, are double or triple those of Whites despite only marginally increased case rates and that these increased death rates generally correspond inversely to pneumococcal vaccination rates. Inspection of Table 4 shows, however, that IPD rates are much better predictors of COVID-19 rates of hospitalization and death than are pneumococcal vaccination rates. Asians have significantly lower rates of invasive pneumococcal disease (IPD) than other minorities despite low rates of pneumococcal vaccination and COVID-19 risks on a par with their low IPD rates. Conversely, Native Americans have very high rates of pneumococcal vaccinations, essentially equivalent to those of Whites, yet they experience extraordinarily high IPD rates, up to five times those of other ethnic groups. The risk of death from COVID-19 among Native Americans is correspondingly about five times as among Whites and Asians (Table 4). Thus, IPD rates are, in this case, as in the Italian case as well, a better predictor of COVID-19 risks than even pneumococcal vaccination rates.

## 4. Discussion

The key findings of this study are as follows. Section 3.2 provided evidence that pneumococcal vaccination rates, particularly the sum of pneumococcal vaccinations across a lifetime, were strongly inversely correlated with COVID-19 case and death rates in a group of 51 nations that had experienced their outbreaks of SARS-CoV-2 at comparable times. A much weaker but statistically significant correlation was also found for a protective effect of MMR vaccination against COVID-19 death. Section 3.3 examined the possibility that pneumococcal vaccinations might interact additively or synergistically with other vaccines to provide even greater protection against COVID-19, but only the addition of influenza vaccination improved COVID-19 protection and only for case rates but not death rates. No other pairwise combinations of vaccines correlated statistically significantly with a decrease in COVID-19 cases or death rates. Section 3.4 demonstrated that pneumococcal vaccination rates were statistically independent of influenza, MMR, and BCG vaccination rates so that these could not be confounding factors in the protection that pneumococcal vaccines appear to convey against COVID-19. Weak correlations were, however, found between lifetime cumulative pneumococcal vaccination rates and those of *Haemophilus influenzae* (Hib) and DTP so that some of the apparent protection against COVID-19 might be influenced by these vaccines. Section 3.5 tested the logical prediction that if pneumococcal vaccination rates are inversely correlated with COVID-19 rates, then rates of invasive pneumococcal disease (IPD) should directly correlate with COVID-19 rates, which was found to be the case. Rates of lower respiratory disease (LRI), however, which is a much broader pulmonary disease category encompassing many other viral and bacterial diseases, did not correlate significantly with COVID-19 rates. Section 3.6 further investigated the IPD-COVID-19 link in the case of the 21 regions of Italy, where, once again, IPD rates accurately predicted rates of death from COVID-19. Notably, rates of influenza vaccination were also found to be inversely correlated with COVID-19 death rates, confirming the possibility that influenza vaccination and pneumococcal vaccination can act additively or synergistically to protect against COVID-19. The protective effect of MMR vaccination found in the international data (Section 3.2) was not verified for the Italian regions. Finally, Section 3.7 demonstrated again the presence of a direct correlation between IPD rates and various measures of COVID-19 risks such as hospitalization and death within the United States. In this instance, IPD rates were much better predictors of COVID-19 risks than were pneumococcal vaccination rates. Once again, as in the international data (Section 3.2), no COVID-19 rates were not associated with lower respiratory infection (LRI) rates. In sum, there is something “special” about pneumococcal infections with regard to COVID-19 risks that is evident in the inverse correlation between pneumococcal vaccination rates and COVID-19 case and death rates, as well as in the direct correlation between invasive pneumococcal disease (IPD) rates and COVID-19 case and death rates. Some evidence exists that influenza and MMR vaccinations may also contribute some level of protection against COVID-19.

One aspect of this “special relationship” between pneumococcal infection and COVID-19 is evident in the fact that patients who contract both IPD and become SARS-CoV-2 positive are four to eight times as likely to die than either IPD patients without a SARS-CoV-2 super-infection or SARS-CoV-2 infection without IPD [166,167]. On this point, some additional data concerning pneumococcal vaccination rates among the elderly in Italy were available but were too sparse to permit statistical analysis. As illustrated in Table 3 and Figure 5 and Figure 6, IPD rates in Italian regions correlate very closely with COVID-19 rates of death, and infant pneumococcal vaccination with PCV13 has only a minor effect on this rate [168,169]. Three factors account for this lack of protection. One is that PCV13 vaccination rates among infants in Italy are very low in comparison with other European and Asian nations (Table 1), so herd immunity is not achieved, and the adult protection from infant immunization observed in other nations (reviewed in [5]) is not achieved in Italy. The second reason is that the pneumococcal serotype distribution responsible for adult IPD in Italy consists almost entirely of types that are not covered by PCV13 [88,166]. The third is that the pneumococcal vaccination rate among adults 65 and over has been described as “extremely low in Italy” [168]. In contrast to Germany, Iceland, Israel, South Korea, Australia, and the U. S., where the over 64 adult pneumococcal vaccination rate is above 50% (Table 1), in the Liguria region of Italy, it was between 26 and 31% [168]; 24.9% in Calabria (“among the highest in Italy”) [170]; 20% in the Friuli Venezia Giulia region [171]; 15.4% in Veneto [172]; and averaged only 5% per year in each of ten years in the Italian region of Puglia, reaching a cumulative total of no more than 24–30% [173]. In consequence, even as infant rates of IPD have been falling in recent years, rates among adults over 64 have been increasing and now account for more than two-thirds of all cases in Italy [87,88,151]. Thus, the lack of protection against IPD afforded by infant PCV13 vaccination in Italy makes perfect sense in the context of the shift of IPD-responsible serotypes to non-covered ones among adults and the very low rates of pneumococcal vaccination among those adults. Given the relationship of IPD to the risk of death following SARS-CoV-2, the high COVID-19 case and death rates observed in Italy could be predicted.

The risks of ethnic minorities in the United States provide additional insights into the relationship between pneumococcal vaccination rates and rates of IPD and COVID-19 that are similarly instructive. Vaccine uptake does not necessarily correlate with efficacy due to other factors that alter immune function, such as nutrition and disease burden, as has been amply documented internationally [174]. This effect is particularly evident in the United States in the mismatch between high pneumococcal vaccination rates among Native Americans and their very high rates of invasive pneumococcal disease (IPD) and COVID-19 (Table 4). In contrast, Native Americans are often vaccinated against pneumococci at rates similar to or even higher than Whites in the United States (Table 4). The efficacy of pneumococcal vaccination has been found to be extremely low [5,159]. The causes of the decreased efficacy of pneumococcal vaccination among Native Americans are multifactorial and involve unusually high rates of endemic disease, malnutrition, obesity, diabetes, smoking, alcoholism, and other effects of poverty [139,157,158,159]. Other U.S. minorities, such as African Americans and Latinos [139,159,160,161,162,163,164,165], also experience lower vaccine efficacy than is found among Whites and for similar reasons. Because poverty, malnutrition, endemic disease, and substance use vary widely by geographical region, significant differences in pneumococcal vaccine efficacy also exist at a global level [174], so that simply using vaccination rates, as has been performed here, and in most of the other studies cited in this paper, is only an approximation for actual efficacy. IPD rates may be better measures of COVID-19 risks precisely because they reflect the actual efficacy of pneumococcal vaccination in any given population.

This study shows that influenza vaccination may have some protective effect against COVID-19 acquisition, though this result was found only as a synergistic effect for the international dataset and for Italy, but not for the United States, and results of other studies have also been contradictory [46,49,50,51,52,53,54,55,61,62,63,64]. Notably, however, a protective effect of influenza vaccination against coronaviruses and respiratory viruses has previously been reported: Wolff [175] found that influenza vaccination in the 2017–2018 influenza season protected against resulted in “virus interference”’ with other respiratory infections, “significantly associated with coronavirus and human metapneumovirus; however, significant protection with vaccination was associated not only with most influenza viruses, but also parainfluenza, RSV, and non-influenza virus coinfections.” [175] Additionally, pneumococcal and influenza vaccinations have been found to synergize in preventing community-acquired pneumonias [139]. Another possibility is that because influenza infection predisposes the lungs to bacterial superinfections with bacteria such as Streptococci and Staphylococci [7,8,9,10,11], influenza vaccination may decrease carriage of such bacteria, thereby indirectly lowering the rate of invasive bacterial disease and thus lowering the risk of severe COVID-19 as well. Further research will be needed to sort the issues out.

The results reported here add to the literature reviewed in the Introduction that cast doubt on studies reporting that BCG, polio, or DPT vaccinations may protect against COVID-19 acquisition or death [64,65,66,67,68,69,70,71,72,73,74,75,76]. However, one important caveat must be mentioned, which is that it was not possible in this study to differentiate between nations using acellular pertussis vaccines, those using whole-cell pertussis vaccines, and those using both as part of their vaccination programs. A significant difference between acellular and whole-cell pertussis vaccinations may very well exist in terms of their protection against COVID-19. Three studies, in particular, can be cited to make this case. One is Sambul et al.ia’s [48] study reporting significant correlations between pertussis antibody titers and protection against COVID-19. This study occurred in Turkey, where the dominant pertussis vaccine is a whole-cell version. Both Root-Bernstein [6] and Reche [77] found that while the acellular pertussis vaccine displayed very few protein similarities with SARS-CoV-2 proteins, the whole-cell pertussis vaccine displayed more than did the proteins of any other vaccine, including the pneumococcal ones. Reche [77] therefore concluded that DPT vaccines containing whole-cell diphtheria vaccine were likely to be the best candidate for protection against COVID-19. However, Root-Bernstein [6] argued that, as a percentage of the protein sequences encoded within the diphtheria genome (0.2 percent), the whole-cell diphtheria vaccine was much less likely to elicit an immune response cross-reactive with SARS-CoV-2 than pneumococcal vaccine proteins, nearly 13% of which mimic the protein sequences of SARS-CoV-2. In the end, the issue comes down to whether the vaccine sequences are, or are not, both major antigens and cross-reactive with SARS-CoV-2, which is a matter for experimental determination.

A similar complication creates difficulties evaluating the effectiveness of MMR as a protective vaccine against COVID-19 since measles-containing vaccines (MCV) do not necessarily incorporate mumps or rubella as components, and thus, the constituents may vary in ways that were not taken into account in this study. The evidence from Sambul et al.’s [48] study that titers specifically against rubella antigens, but not measles or mumps, correlated with protection against COVID-19 suggests that relying on lumped data involving MMR or MCV might be misleading. However, as noted in the Introduction, evidence supporting a role for MMR vaccination as a protective measure against COVID-19 is, at best, controversial.

So, in sum, this study supports previous studies reviewed in the Introduction that suggest a protective role for pneumococcal vaccination against COVID-19 infection and death and establishes the complementary case that invasive pneumococcal disease (IPD) are predictive of increased risks for severe COVID-19. It also provides evidence that while IPD rates are generally inversely correlated with pneumococcal vaccination rates, this relationship does not necessarily hold because of factors that modify vaccine efficacy such as poverty, malnutrition, substance abuse, etc. This study also provides limited support for a role for influenza vaccine, and possibly MMR as well, as additional means of protecting against COVID-19 risks.

### Limitations of This Study

This study has a number of limitations, some of which have just been described in the Discussion section and will not be repeated. Further limitations are discussed here.

The most important limitation of this study is that this is a correlational study, and correlations do not necessarily translate into causation. For example, two quite distinct interpretations are possible of the observation that pneumococcal vaccination rates are generally inversely correlated with IPD rates, while COVID-19 case and death rates are positively correlated with IPD rates. One is that pneumococcal vaccination directly protects against COVID-19 through a mechanism such as increased antibody titers [48] or possibly T-cells [176,177] that are cross-reactive [6] with SARS-CoV-2. This possibility is strengthened by demonstrations that T-cell immunity to SARS-CoV-2 exists in 40–50% of people without known exposure to the virus and that this T-cell immunity is not induced by other coronaviruses [178,179,180,181]. Another possibility is that the pneumococcal vaccination rates are a surrogate measure for IPD rates and that pneumococcal superinfections of SARS-CoV-2 are a major determinant of the risk of severe COVID-19 and death. Taking this interpretation a step further, perhaps IPD rates are themselves merely a surrogate measure for the prevalence of other pneumonia-associated bacteria such as *Mycoplasmas*, *Haemophilus*, and *Klebsiella*, all of which are known to super-infect COVID-19 patients (see Section 2.1 above for discussion and references). Indeed, while 25–30% of elderly pneumonia patients are infected with pneumococci, mycoplasmas, Haemophilus, and Chlamydia infections, each account for another six to eight percent of infections [163]. Thus, COVID-19 risk may be a function of endemic pneumonia-associated disease burden rather than vaccination rate per se.

A second limitation of this study is that only selected nations could be included due to the inability to locate data on both adults over 64 and infant pneumococcal vaccination rates or, in some instances, adult influenza vaccination rates. Moreover, data on at-risk adult pneumococcal vaccination rates were not available for all the nations included in this study, limiting the statistical power of the resulting correlations involving this data. Particularly absent are mainland Asian nations and African nations that, together, represent the majority of the world’s population. Notably, all of the nations included in this study fall into the category of “low incidence, low mortality” rates (<100/1000 incidence and <100/100,000 deaths) of IPD and LRI while most African and Asian nations, and also, notably, Bolivia, are in the “high incidence, high mortality” rates category (>100/1000 incidence and >300/100,000 deaths) of IPD and LRI [81,182]. It is possible, therefore, not only that the results reported here are valid only for industrialized nations but that a completely different epidemiological picture characterizes nations such as India, Pakistan, Bangladesh, and central African nations due to very high endemic pneumococcal and other bacterial disease rates. There are two distinct possibilities that such high bacterial rates might have on COVID-19 epidemiology: one is that high carriage rates and childhood exposure to bacteria such as pneumococci and Haemophilus “naturally” vaccinate the inhabitants, making them resistant to COVID-19; the other is that these high bacterial infection and carriage rates will make the inhabitants of “high incidence, high mortality” nations more susceptible to severe COVID-19. Time will tell.

A third limitation of this study is that the lack of significant correlations between COVID-19 case and death rates and MMR and DTP vaccinations may be misleading. The MMR data were mainly drawn from statistics on measles-containing vaccines. While the majority of these were confirmed to be the MMR vaccine, some may not have been. Since some investigators have identified the rubella component as the most important factor in protection against COVID-19 [48,56,57], it is possible that the weak correlations between MMR and COVID-19 rates found here are due to the absence of rubella in some of the measles-containing vaccine formulations included here. Similar difficulties call into question the lack of DTP-COVID-19 correlations reported here since some DTP vaccines use whole-cell pertussis formulations and others acellular (see Discussion above).

## 5. Conclusions

The data produced here support the hypothesis that vaccination rates and endemic pneumonia burden as measured by invasive pneumococcal disease (IPD) affect population-level measures of susceptibility to serious COVID-19 infections and death. The particular importance of pneumococcal disease measures, and the efficacy of pneumococcal vaccines specifically in moderating COVID-19 risks, strongly suggest that COVID-19, such as pandemic influenza and its associated pneumonias, is the result of coinfections with viruses and bacteria, in this case, SARS-CoV-2 with pneumococci and perhaps other pneumonia-associated bacteria such as mycoplasmas. *Chlamydia, Haemophilus* and *Klebsiella*. The apparent synergy between pneumococcal and influenza vaccinations in moderating COVID-19 cases suggests that SARS-CoV-2 may synergize with influenza A viruses to enhance the severity of disease or that some influenza strains may, by predisposing to bacterial superinfections, increase the severity of SARS-CoV-2. These results argue for increasing pneumococcal, influenza, and possibly MMR vaccination rates as a way of decreasing population-level endemic disease burdens and thus risk of COVID-19 severe disease and death. Such a strategy would have broader and more obvious benefits in protecting the population against these specific diseases as well as against COVID-19. One caveat is that immune dysfunction associated with marginalization and poverty can interfere with vaccine efficacy and make it imperative that more effective means of vaccinating at-risk populations are needed.

## Figures and Tables

**Figure 1 vaccines-09-00474-f001:**
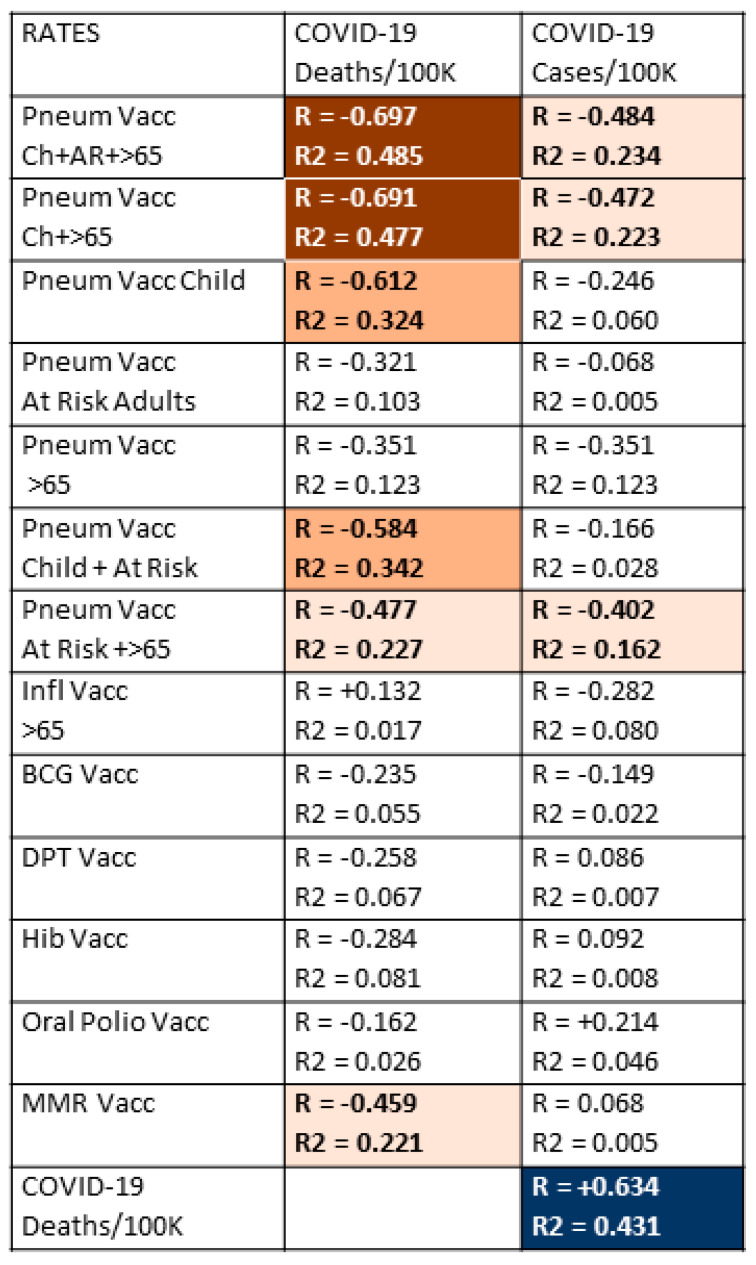
Correlations between the national rates of vaccinations (from Table 1) and national COVID-19 deaths/million population and cases/million population. R is Pearson’s correlation coefficient, providing a measure of how well correlated the two sets of data are; R^2^ is the correlation coefficient, providing a measure of the effect size, or how much of the phenomenon can be attributed to the correlation. The strength of the correlation (R) is indicated by the intensity of the color while the color itself indicates whether the correlation is positive or negative. Dark red blocks with white numbers are the strongest negative correlations in which R is significant at *p* < 0.0005 and R^2^ is significant at *p* < 0.05; orange blocks with black lettering are moderately negative correlations in which R is significant at *p* < 0.005, but R^2^ is not significant; light orange blocks with black lettering are weakly negative correlations in which only R is significant at *p* < 0.05; dark blue blocks with white lettering are strongly positive correlations in which R is significant at *p* < 0.0005 and R^2^ is significant at *p* < 0.05. and finally, white blocks with black numbers are correlations in which neither R nor R^2^ are statistically significant after Bonferroni correction. With 51 samples and after Bonferroni correction for 26 correlations, R values must be 0.392 or greater to achieve a *p*-value of less than 0.05. After Bonferroni corrections, R values of 0.47 or above have a *p*-value less than 0.005, and R values of 0.60 and above have *p*-values less than 0.00005. After correcting for 13 variables, R^2^ must have a value of 0.41 or greater to achieve a *p*-value less than 0.05. In sum, the correlations with R values above 0.4 are highly statistically significant, and the effect sizes as determined by R^2^ reach statistical significance when the values also exceed 0.4. Pneum Vacc = pneumococcal vaccination; Ch = childhood vaccination rate; AR = vaccination rate of at-risk adults 18–64; >65 = vaccination rate of adults 65 years of age and older; Inf = influenza vaccination rate; BCG = Bacillus Calmette–Guerin (tuberculosis) vaccination rate; DPT = diphtheria-pertussis-tetanus vaccination rate; Hib = Haemophilus influenzae type B vaccination rate; MMR = measles-mumps-rubella vaccination rate; vacc = vaccination; /Million = per million population; /1000 = per one thousand population.

**Figure 2 vaccines-09-00474-f002:**
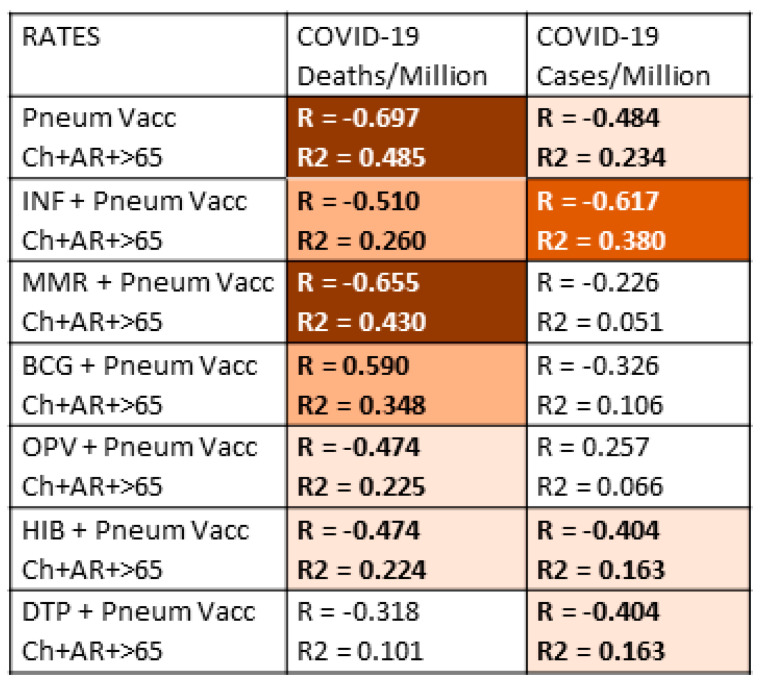
Investigation of whether vaccine combinations produce an enhanced protective effect (as measured by an inverse correlation to COVID-19 case and death rates) of the total pneumococcal vaccination rate. The vast majority of combinations resulted in a decrease in the correlation values indicating that there is no benefit from combining vaccinations; instead, there is an averaging of the vaccine effects. One notable exception was that a combination of pneumococcal and influenza vaccinations correlated with a significant decrease in COVID-19 cases, suggesting additive protection by the two vaccines. All possible pairwise combinations of the vaccine data were correlated with the COVID-19 case and death rates; all of those pairwise combinations not shown in the Table failed to reach significance in both the R and R^2^ values and were therefore omitted for simplicity. The strength of the correlation (R) is indicated by the intensity of the color, while the color itself indicates whether the correlation is positive or negative. Dark red blocks with white numbers are the strongest negative correlations; red blocks with white lettering are less strong negative correlations; orange blocks with bold black lettering are moderately negative correlations; light orange blocks with black lettering are weakly negative correlations; white blocks with black numbers are correlations in which neither R nor R^2^ are statistically significant after Bonferroni correction. With 51 national samples and after Bonferroni correction for 7 correlations per set of samples, R values must be greater than 0.35 or greater to achieve a *p*-value of less than 0.05. After Bonferroni corrections, R values of 0.41 or above have a *p*-value less than 0.005, and R values of 0.51 and above have *p*-values less than 0.00005. After correcting for seven variables, R^2^ must have a value of 0.26 or greater to achieve a *p*-value less than 0.05. In sum, the correlations with R values above 0.4 are highly statistically significant, but the effect sizes as determined by R^2^ reach statistical significance when the values also exceed 0.26. Pneum Vacc = pneumococcal vaccination; Ch = childhood vaccination rate; AR = vaccination rate of at-risk adults 18–64; >65= vaccination rate of adults 65 years of age and older; Inf = influenza vaccination rate; BCG = Bacillus Calmette–Guerin (tuberculosis) vaccination rate; DPT = diphtheria-pertussis-tetanus vaccination rate; Hib = Haemophilus influenzae type B vaccination rate; MMR = measles-mumps-rubella vaccination rate; vacc = vaccination; /Million = per million population; /1000 = per one thousand population.

**Figure 3 vaccines-09-00474-f003:**
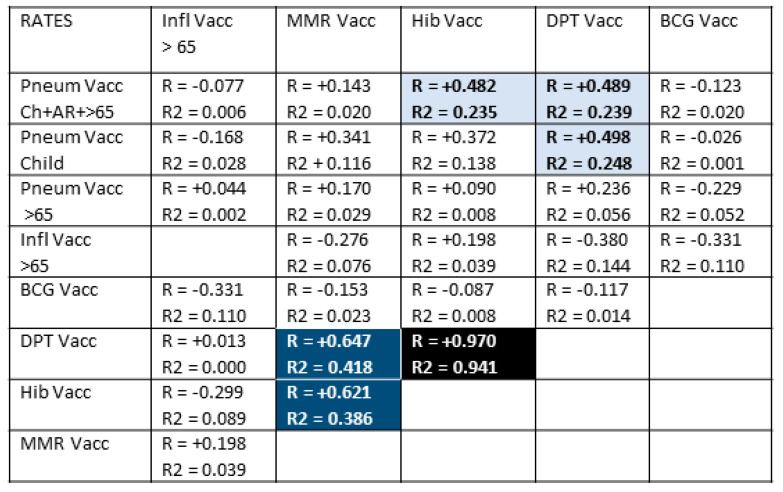
Correlations between rates of vaccinations testing whether confounding effects may account for some of the effects reported in Figure 1 and Figure 2. With 51 samples and after Bonferroni correction for 40 correlations, R values must be greater than 0.40 or greater to achieve a *p*-value of less than 0.05. After Bonferroni corrections, R values of 0.50 or above have a *p*-value less than 0.005, and R values of 0.60 and above have *p*-values less than 0.0005. After correcting for 8 variables, R^2^ must have a value of 0.26 or greater to achieve a *p*-value less than 0.05 and 0.39 to achieve a *p*-value of less than 0.005. While some vaccines are provided at very similar rates and often at the same times (blue and black blocks), there is no significant correlation for the majority of vaccination rates (white blocks). In terms of the additive effect of pneumococcal and influenza vaccination reported in Figure 2, there is notably no significant correlation between rates of influenza vaccination and pneumococcal vaccination at any age, suggesting that these are independent variables. The strength of the correlation (R) is indicated by the intensity of the color, while the color itself indicates whether the correlation is positive or negative. Light blue blocks with black lettering have significant R value correlations; dark blue blocks with white lettering are strongly positive R and R^2^ correlations; black block with white lettering (Hib and DPT vaccination rates) is almost perfectly correlated. White blocks with black numbers are correlations in which neither R nor R^2^ are statistically significant after Bonferroni correction. Pneum Vacc = pneumococcal vaccination; Ch = childhood vaccination rate; AR = vaccination rate of at-risk adults 18–64; >65= vaccination rate of adults 65 years of age and older; Inf = influenza vaccination rate; BCG = Bacillus Calmette–Guerin (tuberculosis) vaccination rate; DPT = diphtheria-pertussis-tetanus vaccination rate; Hib = Haemophilus influenzae type B vaccination rate; MMR = measles-mumps-rubella vaccination rate; vacc = vaccination; /Million = per million population; /1000 = per one thousand population.

**Figure 4 vaccines-09-00474-f004:**
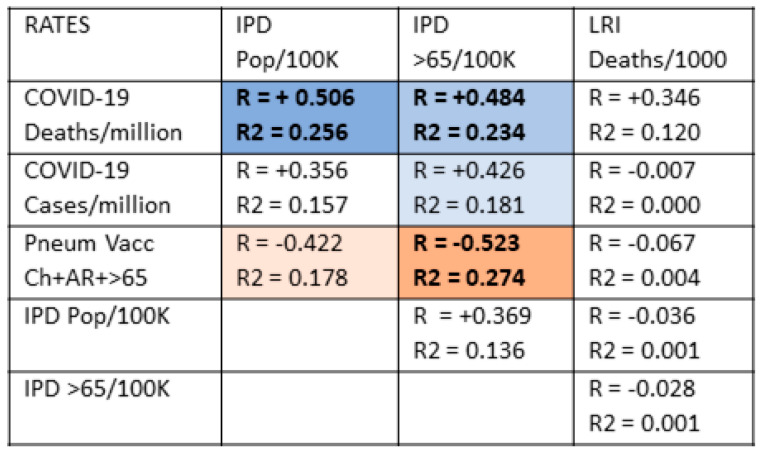
Correlations between COVID-19 case and death rates, invasive pneumococcal diseases (IPD) rates, lower respiratory disease (LRI) rates, and pneumococcal vaccination rates. With 51 samples and after Bonferroni correction for 12 correlations, R values must be 0.40 or greater to achieve a *p*-value of less than 0.05. After correcting for 5 variables, R^2^ must have a value of 0.21 or greater to achieve a *p*-value less than 0.05. Rates of COVID-19 deaths are significantly correlated with IPD rates but not with LRI rates. Conversely, pneumococcal vaccination rates are significantly correlated with decreased IPD rates but have no effect on LRI rates. LRI and IPD rates show no significant correlations. The strength of the correlation (R) is indicated by the intensity of the color, while the color itself indicates whether the correlation is positive or negative. Orange blocks with black lettering are significant negative correlations for both R and R^2^; light orange blocks with black lettering are significant correlations for R but not R^2^; light blue blocks with black lettering are significant positive correlations for R but not R^2^; medium blue blocks with bolded black lettering are statistically significant positive correlations for both R and R^2^. White blocks with black numbers are correlations that are not statistically significant for neither R nor R^2^. Pneum Vacc = pneumococcal vaccination; Ch = childhood vaccination rate; AR = vaccination rate of at-risk adults 18–64; >65= vaccination rate of adults 65 years of age and older; vacc = vaccination; /Million = per million population; /1000 = per one thousand population; pop/100K = rate per 100,000 population.

**Figure 5 vaccines-09-00474-f005:**
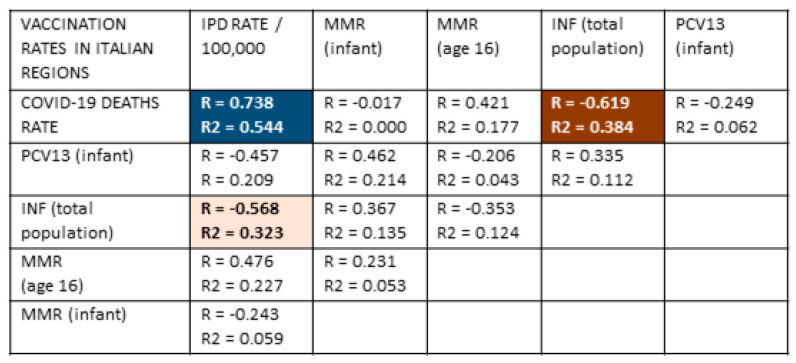
Correlation data comparing vaccination rates for measles-mumps-rubella (MMR) among infants and 16-year-olds, influenza (INF) and pneumococcal vaccine (PPV13) among infants with COVID-19 rates of death (per 100,000 population) and the rate of invasive pneumococcal disease (IPD) in regions of Italy (from Table 3) [50,88,89,143,144,149]. With 21 samples and after Bonferroni correction for 25 correlations, R values must be 0.59 or greater to achieve a *p*-value of less than 0.05. After Bonferroni corrections, R values of 0.69 or above have a *p*-value less than 0.005. After correcting for five variables, R^2^ must have a value of 0.22 or greater to achieve a *p*-value less than 0.05 and 0.30 to achieve a *p*-value of less than 0.005. A very significant positive correlation (both R and R^2^) was found between the IPD rate and COVID-19 death rate and a very significant negative correlation (both R and R^2^) between the influenza vaccination rate and the COVID-19 death rate. The influenza vaccination rate was also significantly inversely correlated (R and R^2^) with the IPD rate. No other correlations were significant for either R or R^2^. The strength of the correlation (R) is indicated by the intensity of the color, while the color itself indicates whether the correlation is positive or negative. The dark red block with white numbers represents the strongest negative correlations; the light orange block with black lettering represents moderate negative correlations; the dark blue block with white lettering represents strongly positive correlations. White blocks with black numbers are correlations that are not statistically significant by either R or R^2^. PCV13 = Pneumococcal conjugate vaccine, 13 strain; Inf = influenza vaccination rate; MMR = measles-mumps-rubella vaccination rate; vacc = vaccination; /Million = per million population.

**Figure 6 vaccines-09-00474-f006:**
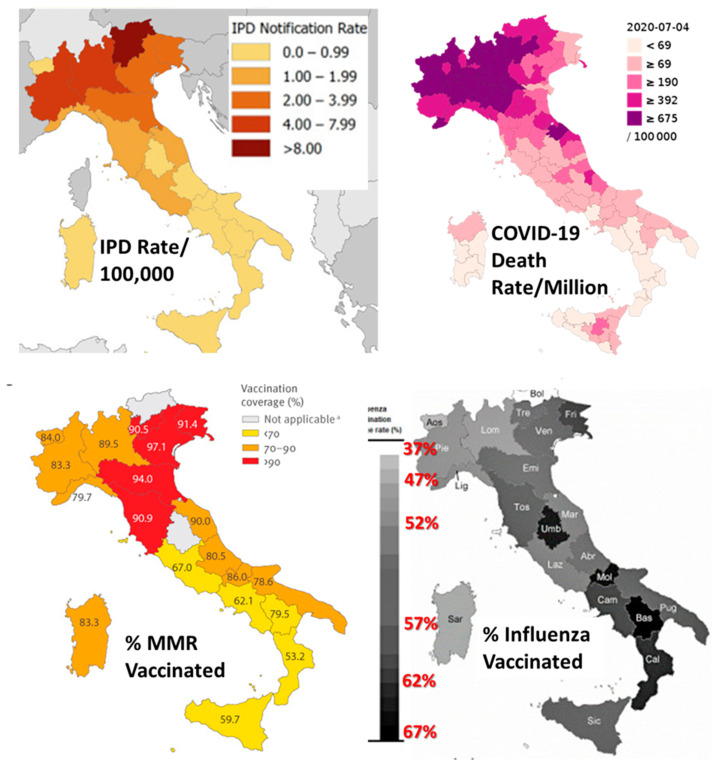
Maps of Italy illustrated the data provided in Table 3 and analyzed statistically in Figure 5. A clear positive relationship is apparent between IPD rates and COVID-19 death rates, as is a clear negative relationship between rates of influenza vaccination and COVID-19 death rates. MMR (measles-mumps-rubella) vaccination rates do not correlate significantly with any of the other factors (see Figure 5). Images modified and assembled from [50,88,89,150,151,152].

**Figure 7 vaccines-09-00474-f007:**
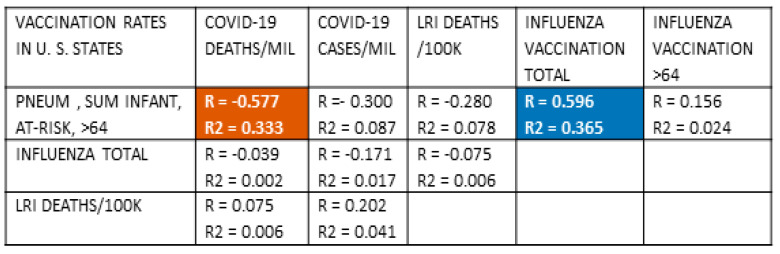
Correlations between COVID-19 death and case rates in the 50 states of the United States of America and three major cities (Chicago, Illinois; Philadelphia, Pennsylvania; and Washington, District of Columbia) with pneumococcal and influenza vaccination rates and rates of lower respiratory infections. With 54 samples and after Bonferroni correction for 10 correlations per set of samples, R values must be 0.39 or greater to achieve a *p*-value of less than 0.05 and 0.49 or greater to achieve a *p*-value of less than 0.005. After correcting for multiple regressions, R^2^ must have a value of 0.27 or greater to achieve a *p*-value less than 0.05. As in previous Figure 1, Figure 4, and Figure 5, significant inverse correlations exist between pneumococcal vaccination rates and COVID-19 death rates. No significant correlations were found, once again (Figure 3), between lower respiratory infection (LRI) rates and pneumococcal vaccination rates or risk of COVID-19 death. Influenza vaccination rates correlated significantly and positively with pneumococcal vaccination rates suggesting the possibility of an additive effect as was evident in the international (Figure 2) and Italian (Figure 5) data. Dark red blocks with white numbers are statistically significant (R and R^2^) negative correlations; dark blue blocks with white lettering are significantly significant (R and R^2^) positive correlations. White blocks with black numbers are correlations that are not statistically significant in either R or R^2^. Pneum Vacc = pneumococcal vaccination; Ch = childhood vaccination rate; AR = vaccination rate of at-risk adults 18–64; >65 = vaccination rate of adults 65 years of age and older; Inf = influenza vaccination rate; vacc = vaccination; /Mil = per million population; /100K = per one hundred thousand population.

**Table 1 vaccines-09-00474-t001:** Vaccination rates with pneumococcal vaccines (Pn) within various age groups (Ch = Childhood; Ad = Adult; AR = At Risk Adults; see text for vaccine descriptions); vaccination with *Hemophilus influenzae* type B vaccine (Hib); and influenza vaccine (Inf); Bacillus Calmette–Guerin (BCG or tuberculosis) vaccine; oral polio vaccine (OPV); diphtheria-tetanus-pertussis (DTP) vaccine; and measle-containing vaccines (MCV).

Nation	COVID Cases / mil	COVID Deaths /mil	LRI Deaths /1000	IPD/ 100K	Ch Pn	Ad Pn	AR Pn	Ch+ Ad Pn	All Pn	Hib	Ad Inf	DTP	M C V	O P V	BCG
Andorra	82,144	983	47.7		96	0	0	96	96	98		99	99	99	0
Argentina	26,842	723	62.8	36	80	18.5	37	99	136	83		83	86	83	93
Australia	1080	35	18.4	8	95	80	36	175	211	94	80.5	95	93	95	0
Austria	15,400	148	8.8	6.9	62	50	10	112	122	85	75	85	94	85	0
Bahrain	48,091	191	6.3		97	0	31	97	128	99	18	20	99	99	97
Belgium	41,297	1079	45.4	13.2	[96]48	29	20	77	97	97	63	98	85	98	0
Bolivia	12,125	748	60.3		75	0		75		75		85	79	75	85
Brazil	26,427	760	33	45	[84]64	10	10	74	84	83	78	87	76	85	98
Canada	6702	275	25.1	20	81	37	17	118	135	91	64	91	87	91	0
Chile	27,037	754	27.4	33.9	[93]71	5		76		95	41	95	93	96	96
Croatia	15,210	175	16	0.2	83	77		160		94		94	93	94	98
Czech Rep	36,576	404	27.2	5	71	37		108		97		97	92	97	99
Denmark	9170	127	40.5	13.8	97	65	6	162	168	97	52	97	90	97	0
France	25,440	610	31.8	8	81	18	26	99	128	95	63	96	83	70	22
Germany	7646	135	27.3	10	86	50.9	16	137	153	92	45.5	96	93	93	0
Greece	5022	69	24.6	0.4	96	34	30	130	160	99	70	99	83	99	45.5
Hong Kong	713	14			99	40		139			47	99	99.8	0	99.8
Hungary	10,322	233	9.8	3.4	99	36	4	135	139	93		99	99	99	99
Iceland	14,733	53	28.8	9.7	92	50	26	142	168	93		93	93	93	0
Ireland	13,020	391	28.9	11.8	86	36	16	122	138	94	62	94	89	94	43
Israel	34,618	287	20.8	9.6	95	70		165		98	59	98	96	98	0
Italy	14,276	672	15.2	12.1	49	26	12	75	87	94	60	95	89	96	0
Japan	829	14	76.6	2.5	99	74	13	173	186	99	27	99	93	98	99
Kuwait	30,394	187	8.7		97	25	79	122	201	99	17	34	98	99	96
Mexico	7335	725	14.5	7	86	45	30	131	161	88	63	88	99	82	96
Netherlands	23,194	460	42.6	16	94	2		97		93	79	93	83	94	0
New Zeal.	395	5	16.3	10.2	96	50	15	146	161	91	62	92	99	92	10
Nicaragua	841	24	13.5		99	0		99		99		99	99	99	99
Norway	4341	52	33.6	11	94	15		114		99		99	99	97	44
Oman	22,747	250	10.1		99	10	17	109	126	99	17	75	99	99	99
Peru	27,610	1048	54.5	19.8	(84)42	5		47		84	69	84	66	87	90
Poland	13,052	193	26.5	3.6	60	69	9	129	138	95		95	93	87	92
Portugal	16,385	274	60.4	3.9	(98)49	5		59		99	68	99	96	99	21
Qatar	47,657	83	2.5		98	20	48	118	165	98	24	29	99	99	99
San Marino	42,049	1296			58	2	4	60	64	86	21	88	86	88	0
Saudi Arab.	9995	157	14.6	17.4	96	7	74	103	177	96	15	7	96	97	98
Singapore	9895	5	58.8		82	60	41	142	183	96		96	98	96	98
Slovakia	13,018	58	32	1.8	97	17.6		114.6		97	68	97	97	97	90
Slovenia	20,518	239	30.3	12.2	97	6	0	103	103	96		95	94	95	96
South Africa	12,956	354	58.3	50	86	3	13	89	102	85		85	83	98	84
South Korea	530	9	21.9	0.79	97	60	30	157	187	98	74	98	97	85	99
Spain	29,692	830	23.5	56.2	(61)31	33	11	64	80	94	60	93	94	96	0
Sweden	14,471	595	31.1	13.9	97	28		125		97	44	97	95	98	26
Switzerland	24,423	312	21.5	7.6	84	2	59	86	144	95	39	96	89	96	0
Taiwan	26	0.3	37.5	7.2	100	41	21	141	162					99	0
Turkey	4569	126	8	2.4	97	24	20	121	141	98	6	98	87	99	96
UAE	14,085	51	4.7	13.6	99	21	43	120	163	99	22	17	99	95	94
U.K.	16,857	713	61.2	11.4	92	44	15	136	151	94	83.5	94	88	93	0
Uruguay	1287	20	46.5		95	0	46	95	141	94		94	90	93	99
USA	30,206	729	29.7	23	88	62	23	150	173	92	65.2	94	97	93	0
Venezuela	3486	30	15.2		0	0	12	0	12	64	0	64	93	62	91

LRI = lower respiratory infections; IPD = invasive pneumococcal disease; mil = million; 100K = 100,000 population. Data in parentheses () are rates of PCV7 vaccination, which have been halved because they only cover half of the PCV13 streptococcal variants, while data in brackets [] are rates of PCV10 vaccination, which have been similarly divided since they cover only 10/13 of the PCV13 variants. Data are arranged by COVID-19 case rate as in Table 1. Data are limited to nations for which data in all key categories were available. Some potentially useful data were not available from sufficient countries to permit useful comparisons, including infant influenza vaccination rates and Hib vaccination rates in groups other than infants, and so these data are not included here. Note that data from China have been excluded because serious concerns have been reported about the data available at this time; vaccination rate data were often lacking as well.

**Table 2 vaccines-09-00474-t002:** Data concerning the rates of COVID-19 cases and deaths in the 50 states of the United States of America; the rates of pneumococcal vaccination (PNEUM) among children, at-risk adults 18 to 64 years of age, and adults 65 years of age and older; the sum of those pneumococcal vaccination rates; and influenza vaccination (INF) rates among adults 65 years of age and older and among the general population of all ages. Three major cities are also included: Chicago, IL (Illinois); Philadelphia, PA (Pennsylvania); and Washington, District of Columbia. INF = influenza; VACC = vaccination; pop = population; >64 = over the age of 64 years; 18-64 = between the ages of 18 and 64, inclusive; CHILD = childhood vaccination.

USA State	COVID-19 Cases/ Million pop	COVID-19 Deaths/ Million pop	Pneumonia Deaths/ 100,000 (2019)	PNEUM CHILD (%) (2017)	PNEUM 18–64 AT RISK (%) (2017)	PNEUM >65 (%) (2016)	TOTAL PNEUM RATE	INF VACC RATE >64 (2019)	INF VACC RATE TOTAL (2019)
Alabama	41,780	629	21.4	87.4	29.4	71.7	188.5	58.0	45.8
Alaska	25,584	115	12.1	83.9	24.3	64.2	172.4	49.0	42.1
Arizona	35,619	847	12.4	84.4	30.5	74	188.9	50.6	43.2
Arkansas	40,382	691	18	77.5	32.2	75	184.7	47.6	51.7
California	24,694	455	15.6	84.9	29	69.9	183.8	54.9	47.5
Colorado	22,745	416	9.7	84.4	37.3	78.8	200.5	55.9	51.4
Connecticut	21,913	1,310	14.8	89	28.8	72.4	190.2	55.3	56.3
Cook County, IL (Chicago)	19,503	501		87.7	30.1	60.9	178.7		
Delaware	27,320	737	13.1	88.8	39.6	69.3	197.7	58.3	51.6
District of Columbia	25,628	928		88.5	30.2	54.9	173.6	59.8	49.3
Florida	39,292	797	9.6	84.1	25.9	73.3	183.3	52.8	41.8
Georgia	38,245	815	14.5	83.2	27.7	71.2	182.1	52.6	43.0
Hawaii	11,263	156	24.5	82.1	25.2	65.3	172.6	52.7	46.8
Idaho	40,827	384	11.9	87.5	28.3	70.5	186.3	53.0	41.4
Illinois	38,510	832	16.3	88.7	28.3	68.6	185.6	55.4	49.8
Indiana	31,249	688	14	78.9	31.6	72.5	183	46.9	48.0
Iowa	49,703	585	15.7	88.8	33.9	74.1	196.8	59.7	53.8
Kansas	34,082	400	17.1	83.9	30.3	76.2	190.4	56.3	50.0
Kentucky	27,047	350	18.3	92.4	38	64.3	194.7	59.4	48.4
Louisiana	40,432	1298	15.6	79.1	30.2	68.6	177.9	45.7	44.0
Maine	5723	113	15.3	82.2	43.1	77.2	202.5	48.5	53.0
Maryland	25,700	698	13.6	87.6	33.9	74.7	196.2	56.5	53.1
Massachusetts	24,867	1472	15.8	91.5	32	73.4	196.9	54.5	56.8
Michigan	22,930	796	14.5	82.9	36.8	72.4	192.1	51.9	48.3
Minnesota	32,070	481	10	79.6	30.5	72.7	182.8	59.1	53.4
Mississippi	42,741	1157	26.1	92.3	25.2	68.7	186.2	60.0	44.1
Missouri	35,562	542	18.7	83.2	33.1	74.4	190.7	61.2	47.5
Montana	37,126	427	10.7	87.3	33.6	73.3	194.2	57.4	47.7
Nebraska	42,594	363	16.1	89.5	31.5	75.8	196.8	57.9	55.3
Nevada	35,720	601	16	83.9	28.8	68.5	181.2	59.5	42.3
New Hampshire	9184	360	14.3	90	35.3	78.6	203.9	50.7	54.2
New Jersey	29,281	1865	12.6	80.1	25.9	68.1	174.1	63.7	45.2
New Mexico	26,173	533	14.2	86.3	32.5	71.3	190.1	53.2	48.8
New York	29,080	1739	18.4	79.6	26.2	64	169.8	44.8	48.6
New York City	30,953	2738		74.2	34	65.9	174.1		
North Carolina	27,969	439	16.9	87.9	34.2	76.7	198.8	65.6	53.5
North Dakota	71,261	839	15.6	85.3	32.1	75.2	192.6	56.5	52.9
Ohio	21,410	476	15.7	81.6	32	74.3	187.9	55.8	48.6
Oklahoma	34,435	363	17.8	78.1	35.1	75.1	188.3	64.5	52.3
Oregon	11,961	173	10.2	83.3	42.5	77.3	203.1	49.2	48.2
Pennsylvania	18,369	711	15.5	82	32.3	75.3	189.6	59.7	53.0
Philadelphia, PA	30,000	1192		83.2	30.1	67.2	180.5		
Rhode Island	34,341	1155	13.3	88.5	34.4	73	195.9	51.1	56.8
South Carolina	36,065	784	14.5	80.2	27.8	73.6	181.6	58.4	47.7
South Dakota	62,625	606	20.7	82.5	26.5	76.9	185.9	58.4	55.1
Tennessee	41,272	526	20.7	84.2	30.8	74.5	189.5	51.1	45.1
Texas	35,177	665	12.9	85.5	26.7	71.3	183.5	48.1	42.2
Utah	41,367	206	13.4	83.8	32.5	73.7	190	45.2	48.7
Vermont	3830	95	9.8	86.5	43.7	74.8	205	52.0	54.5
Virginia	22,667	435	13.1	90.2	38.5	73	201.7	54.4	55.7
Washington	15,856	322	10.9	84.7	41.6	78.1	204.4	58.9	53.4
West Virginia	15,849	280	21.2	86.7	34.7	73	194.4	58.4	50.9
Wisconsin	45,928	397	14.1	87.9	31.2	74.3	193.4	63.5	56.6
Wyoming	29,909	197	18.9	85.8	27.9	70.1	183.8	45.4	43.8

**Table 3 vaccines-09-00474-t003:** Data for the regions of Italy regarding the rates of COVID-19 deaths per 100,000 (100K) population; invasive pneumococcal disease (IPD) per 100,000 (100K) population; percent of infants vaccinated with the PCV13 pneumococcal vaccine (PCV13); percent of infants vaccinated with measles-mumps-rubella vaccine (MMR); percent of 16 years old vaccinated with MMR; and the percent of the population vaccinated against influenza (INF).

ITALIAN REGION	COVID-19 DEATHS/ 100K	IPD/ 100K	PCV13 % (INFANT2016)	MMR % (Infant)	MMR % (Age 16)	INF % (Total Pop.)
ABRUZZO	400	1.0	89.3	92.2	80.5	56
AOSTA VALLEY	800	2.0	87.4	87.8	84.0	45
BASILICATA	70	0.2	97.0	90.2	79.5	67
BOLZANO	500	9.0	80.5	70.8	91.0	37
CALABRIA	35	0.2	90.0	85.4	53.2	64
CAMPANIA	70	0.2	82.1	86.9	62.1	62
EMILIA ROMAGNA	400	3.0	90.6	93.9	94.0	56
FRIULI V. G.	50	2.0	81.4	91.4	90.4	60
LAZIO	130	1.5	93.8	89.6	67.0	53
LIGURIA	500	1.5	91.8	87.9	79.7	51
LOMBARDY	1000	8.0	85.7	94.8	84.4	49
MARCHE	600	2.5	89.4	92.4	90.0	53
MOISE	130	0.5	91.5	88.8	67.0	66
PIEDMONT	700	7.1	91.8	93.1	83.3	50
PUGLIA	70	0.2	91.4	92.3	78.6	58
SARDINIA	70	0.2	94.2	95.5	83.3	47
SICILY	150	0.5	91.7	86.8	59.7	56
TOSCANY	250	6.2	89.0	92.7	90.9	58
TRENTO	700	8.7	87.3	88.5	90.5	55
UMBRIA	130	0.5	94.3	95.2	89.0	65
VENETO	300	4.0	86.6	93.0	97.1	56

**Table 4 vaccines-09-00474-t004:** Rates of pneumococcal vaccination and invasive pneumococcal disease (IPD) among adults >64 in the United States by race/ethnicity compared with ratios of COVID-19 case rates, hospitalizations, and deaths. Data from [159,160,161,162,163,164,165]. (*) Note that it has been established that the efficacy of pneumococcal vaccination among Native Americans is dramatically lower than among Whites [157,158], which explains the extraordinarily high IPD rate among Native Americans. Vaccine efficacy is also lower among U.S. minorities than among Whites [159].

Race/Ethnicity	% Adult >64 Pneumococcal Vaccination (2015–2016)	IPD Rate in >64 Adults/100,000 Population	IPD Rate in >64 Adults Compared with Whites	COVID-19 Case Rate Compared with Whites	COVID-19 Hospitalization Rate Compared with Whites	COVID-19 Deaths Compared with Whites
White	68.2–71.0	4.9–16.7	1.0	1.0	1.0	1.0
Asians	49.0–52.6	5.9	1.0	0.7	1.0	1.0
African American/Black	50.2–55.5	11.3–29.4	1.8–2.3	1.1	2.9	1.9
Hispanic/Latino	41.7–48.6	9.2–22.2	1.9–2.5	1.3	3.1	2.3
Native Americans	60.3–64.2 (*)	80–120	3.0–5.0	1.7	3.7	2.4

## Data Availability

“Data not shown” are available from the author upon request.

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
