# Peer review of "Pneumococcal and Influenza Vaccination Rates and Pneumococcal Invasive Disease Rates Set Geographical and Ethnic Population Susceptibility to Serious COVID-19 Cases and Deaths"

_vaccines, 2021, doi:10.3390/vaccines9050474_

Round 1

Reviewer 1 Report

The manuscript by Robert Root-Bernstein, is in an area of interest to the journal and deals with pneumococcal and Influenza Vaccination Rates and Pneumococcal Invasive Disease Rates Set Geographical and Ethnic Population Susceptibility to Serious COVID-19 Cases and Deaths. The investigator presents a huge amount of data, most of which seem fine, in 11 multi-paneled figures and 4 tables!  The manuscript itself is not an easy ready, because of the incredibly detailed data presented, and in many cases not enough background information to explain enough "lead-ins" to the data itself. Hence, the presentation can be vastly improved so that others will take the time to digest all the data presented, and not just to rely on the abstract.

Specific Comments follow. 

Specific Comments: 

1. There are numerous abbreviations, not all of which are defined. Providing a list of abbreviations and their meanings would help. 

2. Because of the vast amount of data presented, it is hard to go through the paper to grasp all that the author is trying to say. Try to make clear all the assays/assessments used leading up to the data presentation. Not everyone will be familiar with all those used, and clarification of them would make it easier for the readers to follow. One should be able to read a paper without having to go elsewhere in the paper to understand the data, and interpretations provided. 

3. The author may know what he is showing, but some statements are given without providing enough information for the reader.

4. The Discussion is much too long and can be a review article on its own. Substantially reduce the Discussion and focus on the most important findings and their interpretations. 

  1. Too many Figures.

  1. A graphical summary figure would significantly enhance the paper.

  1. References are many and some are in wrong formatting. Please review carefully each of them.

Sincerely.

Author Response

Reviewer #1

The manuscript by Robert Root-Bernstein, is in an area of interest to the journal and deals with pneumococcal and Influenza Vaccination Rates and Pneumococcal Invasive Disease Rates Set Geographical and Ethnic Population Susceptibility to Serious COVID-19 Cases and Deaths. The investigator presents a huge amount of data, most of which seem fine, in 11 multi-paneled figures and 4 tables!  The manuscript itself is not an easy ready, because of the incredibly detailed data presented, and in many cases not enough background information to explain enough "lead-ins" to the data itself. Hence, the presentation can be vastly improved so that others will take the time to digest all the data presented, and not just to rely on the abstract.

I thank the Reviewer for their recommendations, which have greatly improved the paper! I have completely restructured the manuscript to pare down the focus; provided a new, clear Introduction that lays out the flow of the argument; broken down the Results into small sections that each make one, clear point; and rewritten the Conclusion to follow the logic introduced in the Introduction and carried out in the Results sections.  As a result of this restructuring, many side-points have been jettisoned. I believe that the result is a much more readable and focused paper.

Specific Comments follow. 

Specific Comments: 

1. There are numerous abbreviations, not all of which are defined. Providing a list of abbreviations and their meanings would help. 

I believe that I have caught all of these; because the paper is lengthy, however, I have chosen to repeat some of the more commonly used abbreviations and their meanings so as to avoid confusion.

2. Because of the vast amount of data presented, it is hard to go through the paper to grasp all that the author is trying to say. Try to make clear all the assays/assessments used leading up to the data presentation. Not everyone will be familiar with all those used, and clarification of them would make it easier for the readers to follow. One should be able to read a paper without having to go elsewhere in the paper to understand the data, and interpretations provided. 

I have restructured the paper to delete many side issues that do not address the main points of the paper, which are that pneumococcal vaccinations protect against COVID-19 and, conversely, rates of invasive pneumococcal disease directly predict rates of COVID-19. The flow of the argument has now been explained in the Introduction, implemented in small, discrete Sections in the Results, and reiterated in the same order and discrete points in the Conclusion.

3. The author may know what he is showing, but some statements are given without providing enough information for the reader.

Hopefully I have taken care of this problem by means of the changes outlined in 2.

4. The Discussion is much too long and can be a review article on its own. Substantially reduce the Discussion and focus on the most important findings and their interpretations. 

 Agreed and done!

  1. Too many Figures.

 I have eliminated all four of the scatter plots. The others are required to provide the necessary controls for the main findings. However, I have pared these down to focus the results on the key findings, thereby making it easier for readers to identify the main point of each Figure.

  1. A graphical summary figure would significantly enhance the paper.

 Provided.

  1. References are many and some are in wrong formatting. Please review carefully each of them.

All references now properly formatted.

Reviewer 2 Report

In this paper the Author verified the hypothesis of the existence of a correlation between pneumococcal vaccination rates and COVID-19 case/death rate. The findings are very persuasive as the study considered data from 51 countries.

The title is informative and trustworthy. Methods are well described and results support the conclusion. 

MINOR COMMENTS

Abstract

  • LRI: specify

Introduction 

  • L59 afte [12] --> )

Methods

  • The number of countries included must be moved to results
  • L267 provide the key words showing (in a figure?) how many papers you exclude and why
  • how do you calculate R2? Did you mean R^2?
  • A Bonferroni correction for correlation is mandatory

Results

  • In general there are too many comments about the findings (e.g. l392-402, 442-449, 511-516 etc). All this comment must be moved to discussion section as in Results you must focus only on findings.

Discussion

- L938 delete "t"

Author Response

Reviewer #2

In this paper the Author verified the hypothesis of the existence of a correlation between pneumococcal vaccination rates and COVID-19 case/death rate. The findings are very persuasive as the study considered data from 51 countries.

The title is informative and trustworthy. Methods are well described and results support the conclusion. 

MINOR COMMENTS

Abstract

  • LRI: specify

DONE!

Introduction 

  • L59 afte [12] --> )

corrected

Methods

  • The number of countries included must be moved to results

DONE

  • L267 provide the key words showing (in a figure?) how many papers you exclude and why

This request in not suitable. This is not a meta-study. I did not attempt to identify ALL possible sources and integrate their results. I used any method I could think of to identify well-validated sources of information adequate for providing the data necessary to “fill in the blanks” in the TABLES so that the statistical comparisons could be run. There is no definable “method” for doing this. It’s a matter of starting with the most obvious sources (WHO websites; peer-reviewed papers; tracking down individual national health websites and their reports and documents; etc.) until the required information was found. In some cases, where there are blanks in the TABLES, I was unable to identify any reputable source, or could only find sources that were more than five years out of date and therefore of questionable application to the current analysis.

  • how do you calculate R2? Did you mean R^2?

Yes, I meant R2 and have corrected this throughout the paper.

  • A Bonferroni correction for correlation is mandatory

The Bonferroni correction has been provided for all of the Tables and the Table colors have all been altered to reflect the resulting changes in which results are still statistically significant after the Bonferroni correction.  I have also added two websites that I used to aid in my Bonferroni corrections as well as the following paper citation:

Curtin F, Schulz P. Multiple correlations and Bonferroni's correction. Biol Psychiatry. 1998 Oct 15;44(8):775-7. doi: 10.1016/s0006-3223(98)00043-2. PMID: 9798082.

Notably, none of the key findings were altered by the Bonferroni corrections!

Results

  • In general there are too many comments about the findings (e.g. l392-402, 442-449, 511-516 etc). All this comment must be moved to discussion section as in Results you must focus only on findings.

I have done my best to carry out this suggestion, limited only by the need to sign-post the meaning of the results in relation to each other and to the overall argument (vis-à-vis the comments of the other Reviewer).

Discussion

- L938 delete "t"

DONE

Round 2

Reviewer 1 Report

No comments. The author addressed my concerns.

Sincerely